# Cholesterol-rich lipid rafts mediate endocytosis as a common pathway for respiratory syncytial virus entry into different host cells

Anqi Zhou,[1,2] Bao Xue,[2,3,4] Jiayi Zhong,[2] Junjun Liu,[2] Ran Peng,[1] Fan Wang,[1] Yuan Zhou,[2] Jielin Tang,[1,2] Qi Yang,[1,2] Xinwen Chen[1,2,3]

**ABSTRACT** The entry of respiratory syncytial virus (RSV) into host cells is a multifaceted process involving viral adsorption, interaction of viral glycoproteins with cellular receptors, and utilization of various invasion pathways. Despite these complexities, our understanding of potential common pathways facilitating RSV entry remains limited. In this study, we demonstrate that endocytosis via cholesterol-rich lipid rafts is a common mechanism utilized by various RSV genotypes in different cell types. Specifically, RSV strains A2, B18537, and the currently epidemic ON1 strains all employ this mechanism to gain entry into human cell lines, including HEp-2, A549, and primary human bronchial epithelial cells. Cellular receptors binding to viral fusion glycoprotein were recruited to cholesterol-rich lipid rafts, leading to actin rearrangement and endosome formation, which facilitated viral entry. Furthermore, reducing cholesterol levels using methyl-β-cyclodextrin, simvastatin, or terbinafine inhibited RSV infection. Notably, combining simvastatin with an RSV fusion protein inhibitor (AK0529) resulted in enhanced antiviral effects both *in vitro* and *in vivo*. These findings expand our understanding of viral-host interaction and provide a novel therapeutic strategy for treating RSV infection.

**IMPORTANCE** Respiratory syncytial virus (RSV) is an important human pathogen that causes severe bronchiolitis and pneumonia in infants and young children. RSV entry host cells involve generally different invasion pathways and are multistep processes. However, our understanding of the associated common pathways for viral entry remains limited. Our study uncovers a pivotal role for cholesterol-rich lipid rafts in facilitating RSV entry across diverse host cells, a finding that advances our understanding of viral-host interactions and paves the way for novel antiviral strategies. By meticulously examining various RSV genotypes, we revealed shared mechanisms underlying viral entry, highlighting the significance of cholesterol regulation and its impact on infection inhibition. Our findings also demonstrate enhanced antiviral efficacy through a combined approach targeting both viral entry and cholesterol metabolism.

**KEYWORDS** respiratory syncytial virus, cholesterol-rich lipid rafts, endocytosis, entry

Causing over 33 million episodes of lower respiratory tract infections and leading to approximately 3.6 million hospitalizations in a year, respiratory syncytial virus (RSV) stands as the predominant respiratory pathogen among infants and young children (1, 2). Furthermore, there has been a notable increase in the incidence of RSV infection over the past few decades (1, 3, 4). In addition to its impact on children, RSV is also implicated in acute exacerbations of chronic obstructive pulmonary disease in older adults, contributing significantly to morbidity and mortality rates (5). Despite this substantial disease burden, there remains an unmet need for approved RSV-specific small-molecule therapeutics.

**Peer Reviewer** Jinsheng He, College of Life Sciences & Bioengineering, Beijing Jiaotong University, Beijing, China

Address correspondence to Xinwen Chen, chen_xinwen@gzlab.ac.cn, Qi Yang, yang_qi@gzlab.ac.cn, or Jielin Tang, tang_jielin@gzlab.ac.cn.

Anqi Zhou and Bao Xue contributed equally to this article. The author order was determined by drawing straws.

The authors declare no conflict of interest.

See the funding table on p. 21.

RSV is characterized as a pleomorphic, enveloped virus with negative-sense, single-stranded RNA. When observed under spherical forms, it measures approximately 60 nm in diameter, which can expand up to 300 nm in filamentous structures. This virus falls under the *Orthopneumovirus* genus within the *Pneumoviridae* family. RSV exists in two primary antigenic subgroups: serotypes A and B (6). Currently, the ON strain of subtype A and the BA strain of subtype B are the most prevalent strains globally (7–9). The F protein plays a crucial role in receptor binding and membrane fusion, making it a key target for eliciting protective immunity and a focal point in vaccine development efforts (10). Notably, the F protein exhibits promiscuous binding capabilities, interacting with a variety of cellular receptors such as nucleolin (NCL) (11), epidermal growth factor receptor (12), insulin-like growth factor-1 receptor (IGF1R) (13), and intercellular adhesion molecule-1 (14). These interactions facilitate RSV entry through mechanisms involving attachment, endocytosis, and subsequent fusion (15).

Enveloped viruses predominantly infiltrate host cells through mechanisms involving membrane fusion and endocytosis, with endocytosis encompassing clathrin-mediated endocytosis, macropinocytosis, and clathrin-independent pathways (16). Clathrin-independent endocytosis notably occurs via cholesterol-enriched lipid rafts, further categorized into caveolae- or glycolipid raft-mediated processes, contingent upon the specific composition of the lipid raft domain (17). Research has elucidated multiple entry routes for RSV across various cell lines and viral strains (18–22). In HEp-2 cells, RSV A2 exploits surface plasma membrane fusion (18), whereas the RSV Long (type A) strain exploits cholesterol-rich lipid rafts and/or actin-associated endocytosis for cellular invasion (19). Conversely, RSV A2 employs clathrin-mediated endocytosis and macropinocytosis to penetrate HeLa cells, whereas caveolae-mediated endocytosis and macropinocytosis are pivotal in A549 cells (20, 21). Furthermore, RSV A2 accesses human bronchial epithelial cells (HBEC) via cholesterol-rich lipid raft-mediated internalization (22). Nevertheless, a unifying entry pathway shared among diverse RSV genotypes that facilitates broad cellular tropism remains elusive.

Lipid rafts represent areas of the cell membrane that show rapid lateral movement within the lipid bilayer. These rafts function as platforms for protein attachment and can be internalized during signal transduction (23). Cholesterol, a highly asymmetric lipid, is an essential constituent of the cell membrane and plays important structural and signaling roles (24). Cholesterol and sphingolipids are key components of lipid rafts. The use of simvastatin (25) and terbinafine (26) to inhibit the activity of the key enzymes in the cholesterol synthesis pathway, 3-hydroxy-3-methylglutaryl-CoA reductase (HMGCR) and squalene epoxidase (SQLE), effectively lowered cellular cholesterol levels. Disruption of lipid rafts has been shown to affect viral entry, including Marburg virus (MARV) (27), influenza A virus (IAV) (28), and SARS-CoV-2 (29). Additionally, it has been reported that SARS-CoV-2 and IAV also depend on cholesterol for viral entry and attachment (30, 31). Recently, it was also reported that cholesterol-rich lysosomes induced by RSV promoted viral replication by blocking autophagy (32). However, our understanding of the role of cholesterol-rich lipid rafts in RSV entry is currently limited.

In the present study, we conducted a comparative analysis of the entry pathways utilized by various RSV strains into three distinct cell types. Our findings consistently revealed that cholesterol-rich lipid raft-mediated endocytosis is an indispensable mechanism across all examined cell line/viral strain combinations. In our experimental setup, cholesterol depletion led to diminished viral entry and infectivity. Furthermore, the combined administration of the fusion inhibitor AK0529 and simvastatin exhibited an enhanced antiviral effect both *in vitro* and *in vivo*. Collectively, these results underscore cholesterol-rich lipid raft-mediated endocytosis as a general pathway during RSV entry, suggesting its potential as a selective target for therapeutic intervention against RSV infection.

## RESULTS

### Endocytosis mediated by cholesterol-rich lipid rafts is the main pathway in RSV host cell entry

Previous research has indicated that different RSV strains might utilize varying mechanisms to infect different cell types. We investigated the impact of 11 virus entry inhibitors on the replication of RSV strains A2, B18537, and the currently epidemic ON1 strain in HEp-2, A549, and HBEC cells (Fig. 1A) (21, 33–41). Inhibitors targeting the viral F protein, palivizumab, ziresovir (AK0529), and presatovir (GS-5806) effectively suppressed infection across all tested cell lines and strains, highlighting their broad-spectrum efficacy. However, umifenovir hydrochloride (Arbidol), a broad-spectrum viral fusion inhibitor, could not inhibit RSV infection.

Our data also revealed that methyl-β-cyclodextrin (MβCD), an inhibitor of lipid raft-mediated endocytosis, and cytochalasin D (CytoD), an inhibitor of actin-associated endocytosis, dose-dependently inhibited the entry of RSV strains A2, B18537, and ON1 into HEp-2, A549, and HBEC cells. Notably, MβCD completely prevented RSV infection at a concentration of 10 mM (Fig. 1A; Fig. S1). These findings suggest a common endocytosis pathway mediated by cholesterol-rich lipid rafts for different RSV genotypes entering various cells. However, neither $NH_4Cl$ nor chlorpromazine (CPZ) affected RSV infection, indicating that RSV entry is pH insensitive and clathrin independent.

We also observed that the Filippin complex specifically inhibited A2 entry into A549 cells, albeit with weak efficacy (Fig. 1A; Fig. S1B). Additionally, there was a significant difference in RSV infection rate between A549 cells expressing structural domain-negative (DN) caveolin-1 or DN dynamin and wild-type cells but not HEp-2 cells, suggesting that A2 utilizes the caveolae-mediated endocytosis pathway (Fig. S2). Ethyl-isopropyl amiloride (EIPA) exhibited a weak inhibitory effect on the entry of A2, B18537, and ON1 into A549 cells (Fig. 1A; Fig. S1B, E and H), and it had minimal influence on the entry of ON1 into HEp-2 and HBEC cells (Fig. 1A; Fig. S1G and I). These findings suggest that RSV may utilize macropinocytosis pathways as an alternative, rather than the primary route, for cellular entry in A549, HEp-2, and HBEC cells. Conversely, nocodazole (Noc) inhibited infection of B18537 and ON1 strains, but not A2, in a gradient effect (Fig. 1A; Fig. S1), suggesting microtubule involvement in RSV infection but exhibiting strain specificity. However, fusion protein inhibitors did not completely block viral entry, even when used at concentrations exceeding 1,000-fold of their $EC_{50}$ (Fig. 1A; Fig. S1). In contrast, inhibition of cholesterol-rich lipid raft-mediated endocytosis could significantly suppress RSV infection across all tested strains (Fig. 1A; Fig. S1). We next evaluated the effect of cholesterol depletion on RSV binding and internalization. We found that none of the cholesterol depletion affected RSV binding but significantly affected internalization (Fig. 1B and C). Overall, our findings suggested that RSV may utilize cholesterol-rich lipid raft-mediated, actin-associated endocytosis to infect various types of cells.

### RSV entry dynamics analysis by the single particle tracking

Next, we investigated the entry dynamics of RSV in HEp-2 and A549 cells. The RSV virion membranes were labeled with a fluorescent lipophilic dye. We visualized the localization of the lipophilic DiD-labeled RSV, CT-B-labeled lipid rafts, and NBD-labeled membrane cholesterol. As shown in Fig. 2; Movies S1 to S4, the RSV-DiD signal moved from the peripheral region of the cell membrane toward the cytoplasm, colocalizing with signals from lipid rafts and cholesterol in the process. Trajectory analysis revealed that RSV trafficking included directional or bidirectional movements, and at times, the virus remained virtually static (Movies S1 to S4). Time-lapse imaging showed that RSV-DiD particles could cross the cell plasma membrane shortly after binding to the cell surface (Movies S1 to S4). These results suggested that RSV might enter HEp-2 and A549 cells via endocytosis rather than direct fusion with the plasma membrane.

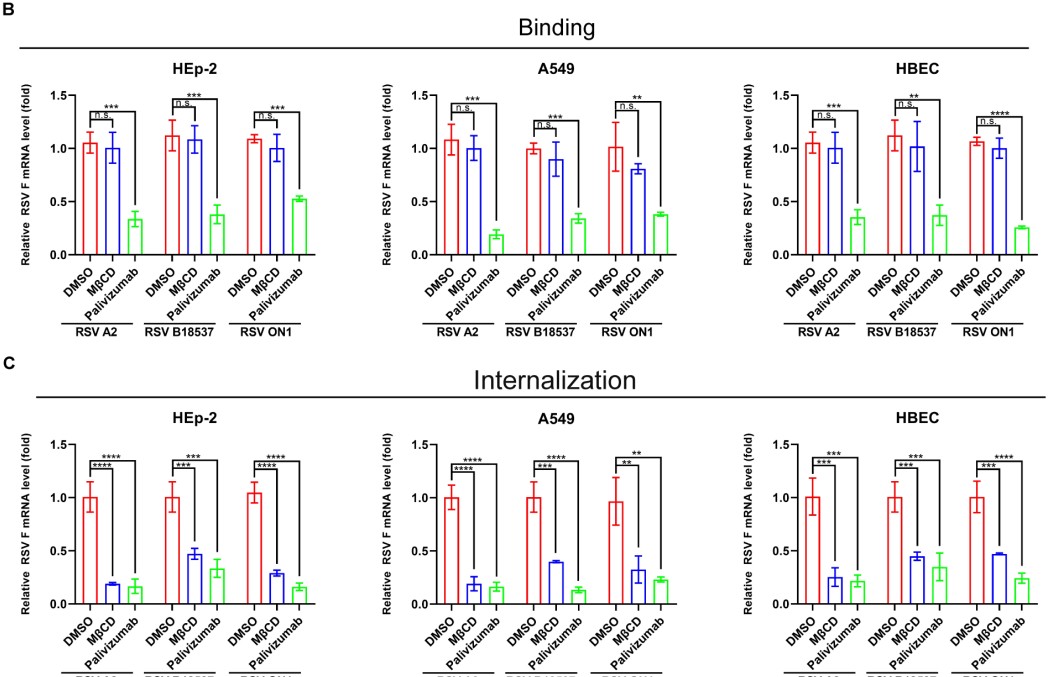

| Inhibitor | HEp-2 | | | A549 | | | HBEC | | | Main pathway |
|---|---|---|---|---|---|---|---|---|---|---|
| Palivizumab | ++ | ++ | +++ | ++ | ++ | +++ | ++ | +++ | +++ | Binding |
| Ziresovir (AK0529) | ++ | +++ | +++ | ++ | +++ | +++ | ++ | ++ | +++ | Fusion |
| Presatovir (GS-5806) | ++ | +++ | +++ | ++ | +++ | +++ | ++ | ++ | +++ | Fusion |
| Umifenovir hydrochloride (Arbidol) | − | − | − | − | − | − | − | − | − | Cell membrane |
| $NH_4Cl$ | − | − | − | − | − | − | − | − | − | Endosome acidification |
| Filipin complex | − | − | − | + | − | − | − | − | − | Caveolae |
| Chlorpromazine (CPZ) | − | − | − | + | − | − | − | − | − | Clathrin |
| Ethyl-isopropyl amiloride (EIPA) | − | − | + | + | + | + | − | − | + | Macropinocytosis ($Na^+/H^+$ exchange) |
| Methyl-beta-cyclodextrin (MβCD) | ++ | +++ | +++ | ++ | +++ | ++ | ++ | +++ | +++ | Lipid rafts (cholesterol) |
| Cytochalasin D (Cyto D) | ++ | ++ | +++ | + | +++ | +++ | ++ | + | ++ | Actin |
| Nocodazole (Noc) | − | ++ | +++ | − | +++ | ++ | − | ++ | +++ | Microtubule |
| **RSV strains** | A2 | B18537 | ON1 | A2 | B18537 | ON1 | A2 | B18537 | ON1 | |

**FIG 1** RSV entry into host cells is primarily dependent on cholesterol-rich lipid raft-mediated endocytosis. (A) The table summarizing results is shown in Fig. S1 from infection experiments with RSV A2, B18537, and ON1 in HEp-2, A549, and HBEC cell lines. *$P < 0.05$ and **$P < 0.01$ are defined as +, ***$P < 0.001$ is defined as ++, ****$P < 0.0001$ is defined as +++; n.s., not significant is defined as −. (B) Cholesterol depletion had no effect on RSV binding. Cells were preincubated in a medium containing methyl-β-cyclodextrin (MβCD; 5 mM) at 37°C for 1 hour. Subsequently, the cells were infected with RSV A2, B18537, or ON1 at a multiplicity of infection (MOI) of 10 on ice for 1 hour, respectively. Palivizumab (10 µg/mL) was used as a control. After five cycles of washing, cells were collected, and the viral RNA was detected. (C) Cholesterol depletion reduces the internalization of RSV into cells. After viral binding, the cells were added to the medium supplemented with MβCD (5 mM) or palivizumab (10 µg/mL), and then transferred to a 37°C incubator for 1 hour. Thereafter, the cells were frozen on ice and then treated with proteinase K (500 ng/mL) on ice for 1 hour. After an additional five washes, the cells were collected for RNA extraction. Data are representative of three independent experiments and are presented as mean ± SD. Statistical differences were determined by one-way analysis of variance (ANOVA) in panels B and C. **$P < 0.01$, ***$P < 0.001$ and ****$P < 0.0001$; n.s., not significant.

## RSV-induced actin rearrangement and endosome formation involve cholesterol-rich lipid rafts

Actin serves as a structural scaffold for the assembly of the endocytotic machinery and is essential for efficient viral invasion through the plasma membrane (42). In this study, we infected HEp-2 and A549 cells with RSV for 90 min and analyzed the changes in actin morphology (Fig. 3A). We conducted imaging analysis on 320 cells per experimental

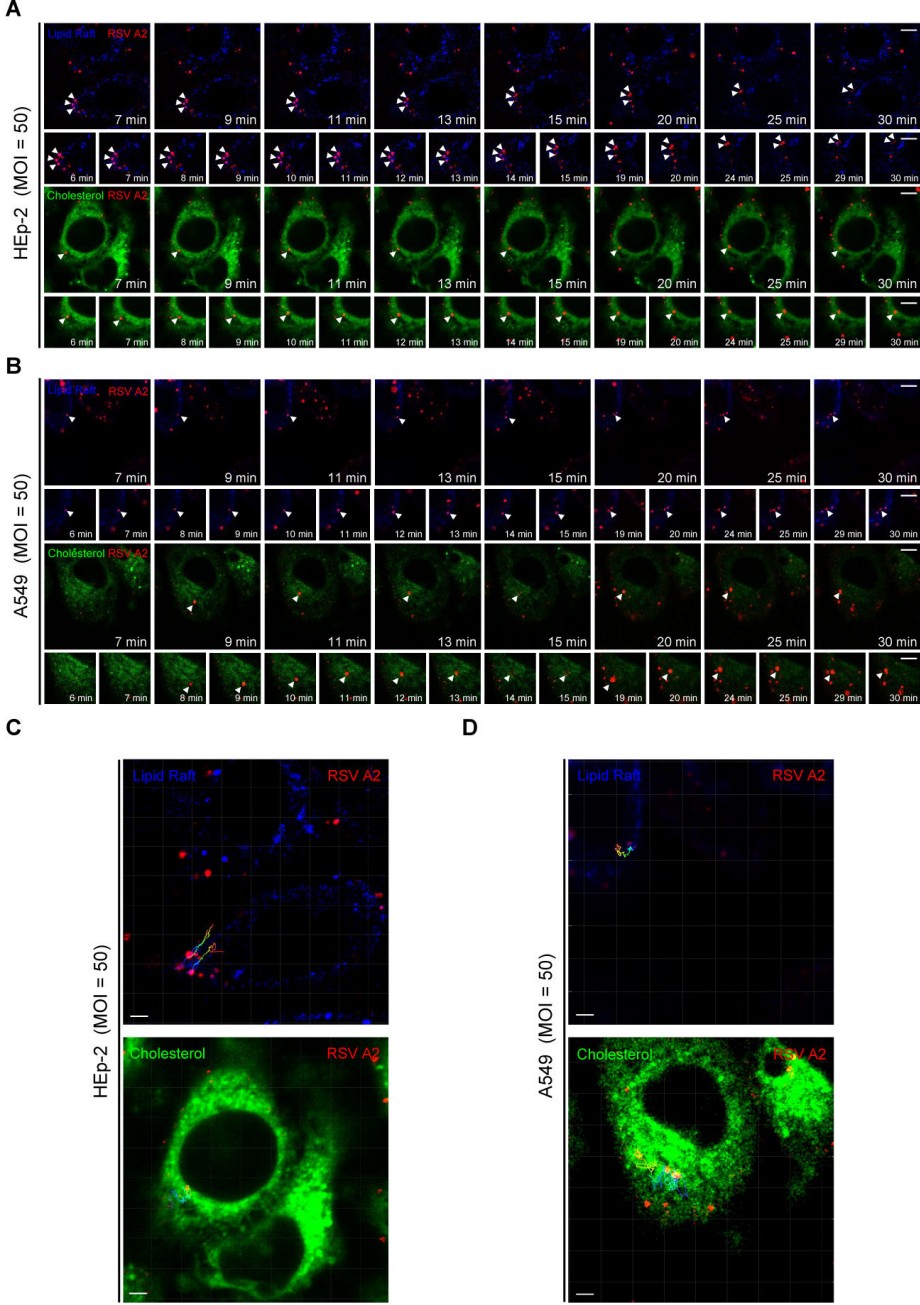

**FIG 2** Ultrastructural and dynamic observations of RSV entry. (A and B) Time-lapse images of RSV-DiD entry. Each image shows the merge signal from RSV-DiD (red) , NBD-cholesterol (green) or cholera toxin subunit B (blue). Scale bar = 10 or 20 µm. (C and D) Trajectories of viruses in panels A and B.

condition to investigate the colocalization of RSV with actin, lipid rafts, and cholesterol in both cell lines. The average number of RSV-associated signals per cell was 5.21 ± 0.61, 3.82 ± 0.35, 8.35 ± 0.79, and 8.13 ± 1.29, respectively. Following a 90 min incubation at 37°C, the RSV signal colocalized with actin and lipid rafts or cholesterol as follows: in HEp-2 cells, 49.13% ± 1.38% of the RSV signal colocalized with actin and lipid rafts, while 32.14% ± 3.52% colocalized with actin and cholesterol. In A549 cells, 25.15% ± 6.23% of the RSV signal colocalized with actin and lipid rafts, and 26.34% ± 5.27% colocalized with actin and cholesterol (Fig. 3A). These findings align with previous studies in NHBE cells (43), which show that RSV infection induces actin rearrangement in HEp-2 and A549 cells, indicating that this phenomenon is likely universal across cell types.

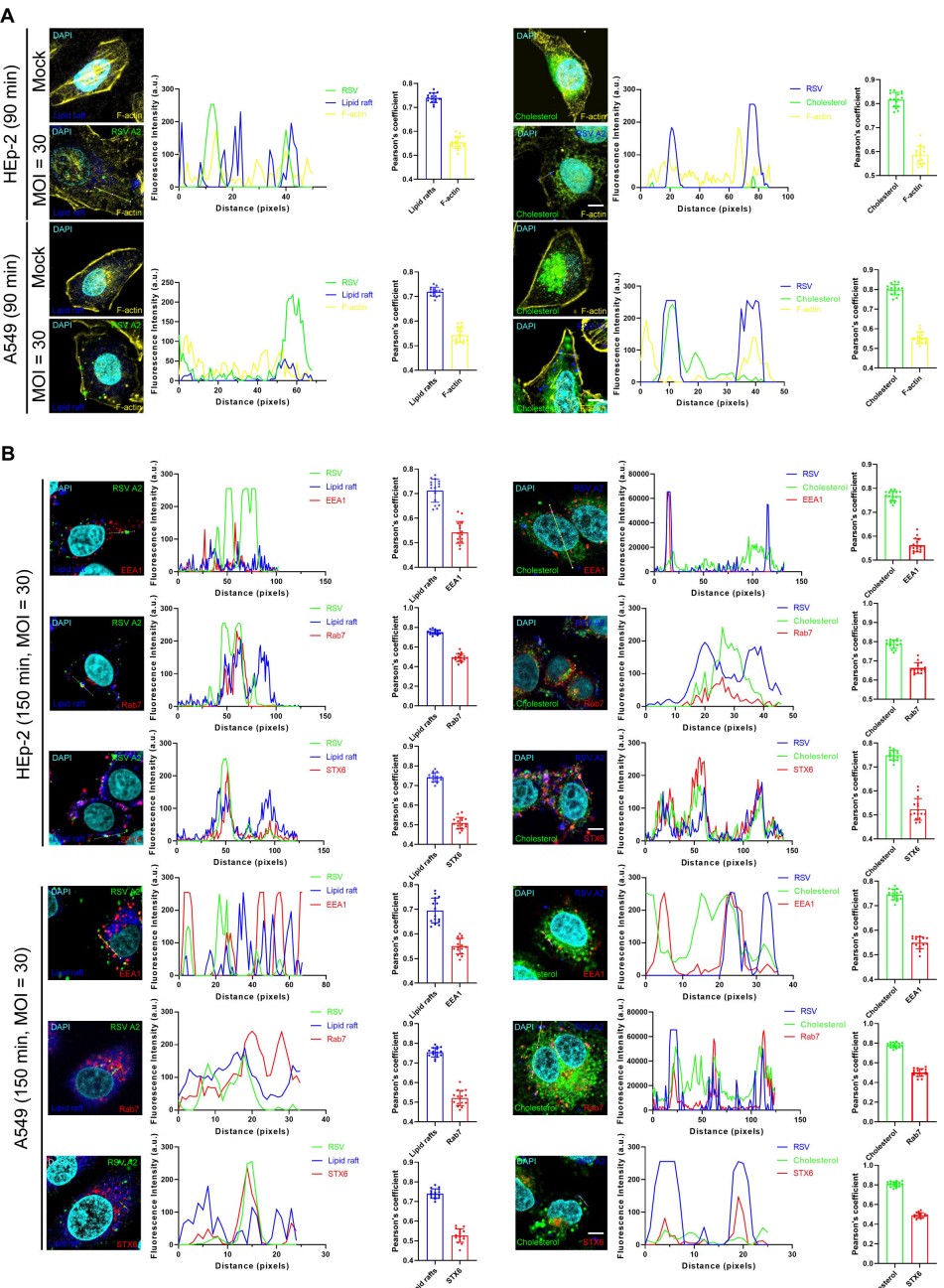

**FIG 3** RSV entry into cells triggers actin polymerization and endosome formation by cholesterol-rich lipid rafts. (A) RSV enters cells by actin-mediated endocytosis. HEp-2 or A549 cells were infected with RSV A2 (MOI = 30) (90 min, 37°C) and fixed (4% PFA, 15 min, room temperature). Cell nuclei were labeled with DAPI (cyan), lipid rafts with Alexa 647-CTB (blue), cholesterol with NBD-cholesterol (green), and F-actin with CellMask orange actin tracking dye (yellow), and images were acquired using confocal microscopy. Scale bar = 10 µm. Intensity profiles show colocalization along the direction of the yellow line. Pearson's correlation between lipid rafts, cholesterol or F-actin, and RSV. n = 16 cells per group. Error bars represent a 95% confidence interval. (B) RSV trafficking from early to late endosomes. HEp-2 or A549 cells were infected with RSV A2 (MOI = 30) (150 min, 37°C) and fixed. Cell nuclei were labeled with DAPI (cyan), lipid rafts with Alexa 647-CTB (blue), and cholesterol with NBD-cholesterol (green). Immunodetection of RSV F (blue), RSV-FITC (green), EEA1, Rab7, and STX 6 (red) was performed using confocal microscopy. Scale bar = 10 µm. Intensity profiles show colocalization along the direction of the yellow line. Pearson's correlation between lipid rafts, cholesterol or F-actin, and RSV. n = 16 cells per group. Error bars represent a 95% confidence interval.

When viruses enter cells via endocytosis, they are typically transported by endocytic vesicles to the early endosomal compartment for sorting. Subsequently, they are transported to fuse with either early or late endosomes (44). In our experiments, we analyzed 320 cells to investigate the colocalization of RSV with early endosomal antigen 1 (EEA1), the late endosome marker Rab7, syntaxin 6 (STX6), and lipid rafts or cholesterol (Fig. 3B). Confocal imaging of HEp-2 cells revealed that the number of RSV-associated signals per cell was 4.21 ± 1.07 for EEA1, 2.23 ± 0.21 for Rab7, 8.52 ± 1.22 for STX6, 4.03 ± 0.79 for lipid rafts, 8.26 ± 1.32 for cholesterol, and 22.63 ± 1.29 for total colocalization events. After a 150 min incubation at 37℃, RSV signals colocalized with endosomal compartments as follows: 51.05% ± 9.24% or 27.43% ± 8.14% with EEA1, 30.3% ± 10.23% or 22.47% ± 6.24% with Rab7, and 20.05% ± 8.32% or 29.39% ± 4.25% with STX6, all in conjunction with lipid rafts or cholesterol (Fig. 3B). In 320 A549 cells, the number of RSV-associated signals per cell was 16.72 ± 0.68 for EEA1, 5.26 ± 2.39 for Rab7, 6.32 ± 3.21 for STX6, 5.26 ± 0.35 for lipid rafts, 15.25 ± 5.26 for cholesterol, and 5.36 ± 1.83 for total colocalization events. Following a 150 min incubation at 37℃, RSV-associated signals colocalized with specific endosomal markers relative to lipid rafts or cholesterol. Specifically, 50.91% ± 3.29% or 25.34% ± 1.47% of RSV signals colocalized with EEA1, 41.76% ± 2.48% or 52.12% ± 4.27% with Rab7, and 33.24% ± 1.74% or 24.30% ± 1.34% with STX6 (Fig. 3B). Although the number of endosomal markers was consistently higher than RSV-associated signals, the distribution of RSV signals corresponded with the progression of endosomal trafficking and maturation, which is consistent with previous reports (43). The above results reveal the critical involvement of lipid rafts and cholesterol throughout the endocytic pathway.

## Receptors are involved in cholesterol-rich lipid raft-mediated endocytosis

First, we examined the distribution of NCL and IGF1R on lipid rafts. While we initially saw very limited localization of NCL and IGF1R to lipid rafts (Fig. 4A and B), infection with RSV significantly increased the accumulation of both molecules on lipid rafts (Fig. 4A and B). Confocal microscopy clearly showed the increased inhomogeneous, mottled staining of distinct membrane areas (Fig. 4C). The number of RSV-associated signals per cell was 10.24 ± 0.98, 3.26 ± 0.27, 9.17 ± 2.34, 20.14 ± 1.62, 10.52 ± 2.46, and 10.23 ± 5.23 in confocal images of 320 HEp-2 cells containing receptors, lipid rafts, or cholesterol. Following a 150 min incubation at 37℃, RSV signals colocalized with lipid rafts or cholesterol at rates of 53.29% ± 1.74% or 55.10% ± 5.14%, respectively. Additionally, RSV signals colocalized with specific receptors as follows: 51.35% ± 4.25% or 33.25% ± 3.14% with NCL, and 77.82% ± 2.45% or 71.61 ± 3.16% with IGF1R, all within the context of colocalization with lipid rafts or cholesterol (Fig. 4C). In confocal images of 320 A549 cells containing receptors, lipid rafts, or cholesterol, the number of RSV-associated signals per cell was 6.17 ± 0.22, 4.14 ± 1.25, 3.15 ± 0.29, 2.15 ± 0.28, 2.26 ± 0.29, and 9.71 ± 1.25. After a 150 min incubation at 37℃, 53.16% ± 0.16% or 51.71% ± 2.16% of RSV signals colocalized with lipid rafts or cholesterol. Regarding specific receptors, 50.19% ± 1.26% or 51.32% ± 2.17% colocalized with NCL, and 66.72% ± 3.16% or 56.26% ± 1.16% colocalized with IGF1R, all within the context of colocalization with lipid rafts or cholesterol (Fig. 4C). These findings indicate that RSV infection induces the aggregation of IGF1R and the subsequent recruitment of NCL from the nucleus to lipid rafts.

It has been reported that SARS-CoV-2 is internalized into the host cell and subsequently traffics to early endosomes through a clathrin-mediated pathway after its binding to the tyrosine-protein kinase receptor UFO (AXL) receptor (45). Similarly, labeled RSV particles were found to colocalize with NCL and IGF1R within the endosomal compartment. The average number of RSV-associated signals per cell was 14.52 ± 1.25, 3.15 ± 0.25, 5.82 ± 1.12, 5.01 ± 0.19, 8.26 ± 0.87, and 4.19 ± 2.15, respectively. After a 60 min incubation at 37℃, RSV-associated signals colocalized with specific endosomal markers relative to NCL or IGF1R. Specifically, 60.25% ± 1.26% and 32.11% ± 2.16% of RSV colocalized with APPL1, 60.22% ± 1.42% and 43.24% ± 2.11% with EEA1, and 52.12% ± 1.62% and 50.01% ± 0.26% with Rab7 (Fig. 4D). The localization of APPL1, EEA1, and Rab7

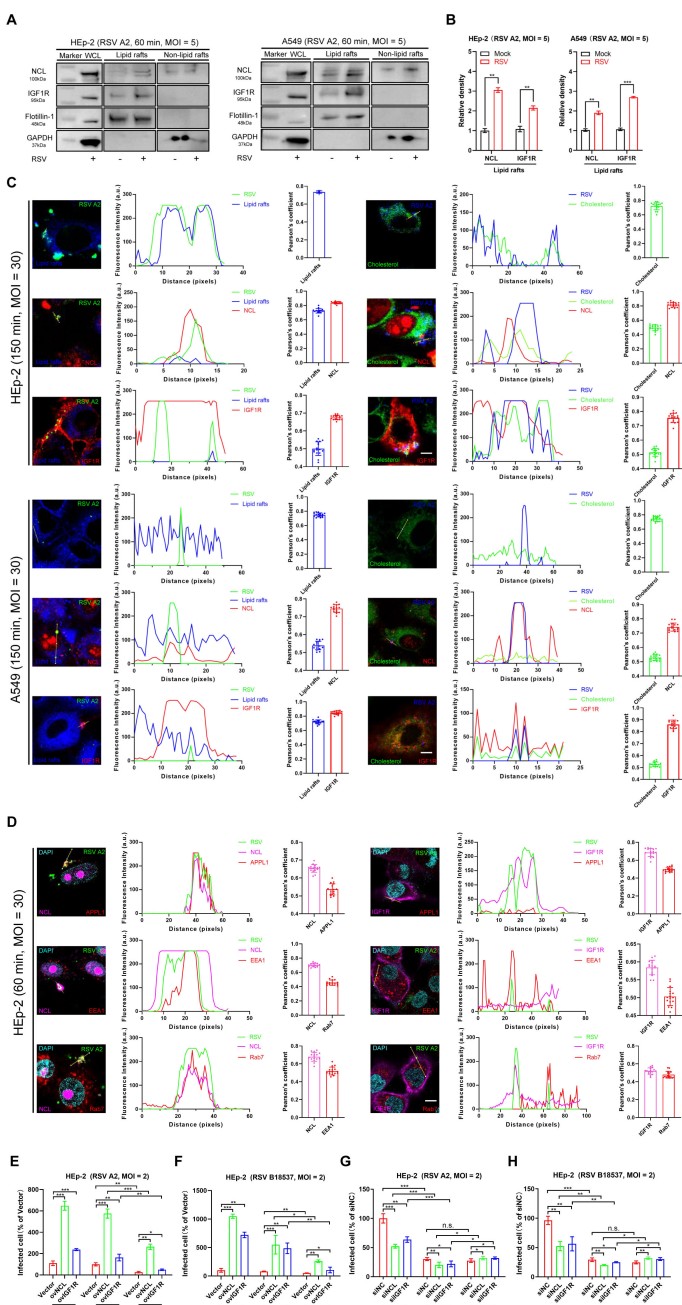

**FIG 4** NCL and IGF1R are involved in cholesterol-rich lipid raft-mediated endocytosis. (A) RSV infection recruited receptors to lipid rafts. HEp-2 or A549 cells were infected with RSV A2 at 37°C for 60 min. Whole cell lysates (WCL) and lipid raft composition were analyzed using western blotting (WB) using anti-Flotillin-1 (lipid raft marker), anti-NCL, anti-IGF1R, and anti-GAPDH antibodies. Representative blots are shown. (B) The relative density of the blots from three independent experiments of the lipid rafts group in A was analyzed by densitometry. (C) RSV co-colocalizes with NCL and IGF1R proteins. HEp-2 or A549 cells were infected with RSV A2 (MOI = 30) (150 min, 37°C) and fixed (4% PFA, 15 min, room temperature). Lipid rafts were labeled with Alexa 647-CTB (blue) and cholesterol with NBD-cholesterol (green). RSV was detected by an anti-RSV F antibody (blue) and anti-RSV-FITC (green), while NCL and IGF1R were detected by corresponding labeled antibodies (red). Scale bar = 10 μm. Intensity profiles show colocalization along the direction of the yellow line. Pearson's correlation between lipid rafts or cholesterol, NCL or IGF1R, and RSV. $n$ = 16 cells per group. Error bars represent a 95% confidence interval. (D) NCL or IGF1R facilitates RSV entry into host cells. HEp-2 cells were infected with RSV A2 (Continued on next page)

**Fig 4 (Continued)**

(MOI = 30) for 60 min and fixed. Antibodies detecting NCL or IGF1R (magenta), RSV (green), and the indicated endocytosis-related proteins (red) were overlaid with cell nucleus DAPI staining (cyan). Scale bar = 10 µm. Intensity profiles show colocalization along the direction of the yellow line. Pearson's correlation between NCL, IGF1R or endocytosis-related proteins, and RSV. $n$ = 16 cells per group. Error bars represent a 95% confidence interval. (E and F) Overexpression of NCL and IGF1R promoted RSV A2 or B18537 (MOI = 2) entry into cells. HEp-2 cells with stable NCL and IGF1R overexpression were pretreated with MβCD (2 mM) for 1 hour, then infected with RSV A2 or B18537 combined with drug treatment for 2 hours. Control cells were treated with an equivalent amount of DMSO alone. Infection rates were quantified using the fluorescence focus assay (FFA) at 24 hours post-infection (h.p.i.). The infection rate was calculated by determining the relative fluorescence intensity compared to the vector group within the untreated condition. Data are representative of three independent experiments and are presented as mean ± SD. Statistical differences were determined by Student's $t$-test. (G and H) Downregulation of NCL and IGF1R genes and MβCD reduced RSV entry into cells. One group of cells was transfected with specific small interfering RNAs (siRNAs) designed to target either NCL or IGF1R for gene knockdown. A second group underwent siRNA-mediated knockdown of NCL or IGF1R, subsequently followed by transfection with plasmids expressing NCL or IGF1R (500 ng/well) to restore their expression levels. Some samples were pretreated with MβCD (2 mM) for 1 hour, while controls were exposed to DMSO alone. Cells were infected with RSV A2 or B18537 (MOI = 2) combined with drug treatment for 2 hours. Infection rates were quantified using the FFA at 24 h.p.i. The infection rate was calculated by determining the relative fluorescence intensity compared to the siNC group within the DMSO-treated condition. Data are representative of three independent experiments and are presented as mean ± SD. Statistical differences were determined by Student's $t$-test. *$P < 0.05$, **$P < 0.01$, and ***$P < 0.001$; n.s., not significant.

in these images suggests that, following binding to IGF1R and recruitment of NCL, internalized RSV fuses with the endosomal membrane to facilitate penetration into the endosomal compartment (Fig. 4D).

To investigate the relationship between RSV receptors and lipid rafts, we evaluated the impact of downregulating or stably overexpressing NCL or IGF1R on the infection of RSV A2 and B18537 strains in HEp-2 cells. Overexpression of NCL or IGF1R increased the infectivity of RSV A2 (Fig. 4E). MβCD treatment reduced the infectivity of RSV A2 but did not affect the increased viral replication caused by the overexpression of receptors (Fig. 4E). Experiments using RSV B18537 confirmed the observations (Fig. 4F).

We also performed receptor silencing and expression restoration experiments. Treating HEp-2 cells with NCL or IGF1R-specific small interfering RNA (siRNA) reduced the expression of the corresponding proteins to 40% and 25% of control levels, respectively (Fig. S3). This reduction was accompanied by a decrease in infectivity in the silenced cells, supporting the involvement of NCL and IGF1R in RSV infectivity. The MβCD treatment causes further decrease for siNCL and siIGF1R in RSV A2 and B18537-infected HEp-2 cells (Fig. 4G and H), but these decreases were slight. The restoration of NCL or IGF1R expression did not significantly increase infectivity, likely because MβCD disrupts lipid raft structures, preventing the normal recruitment of receptors to these rafts for utilization during RSV infection (Fig. 4G and H). These results suggested that receptors were involved in RSV entry into host cells via cholesterol-rich lipid raft-mediated endocytosis.

## Cholesterol depletion effectively inhibits RSV infectivity

To further validate the role of cholesterol in RSV invasion, we analyzed the effects of two rate-limiting enzymes in the cholesterol synthesis pathway (Fig. S4A), specifically HMGCR and SQLE, on RSV infection. Consistent with previous reports, HMGCR knockdown significantly inhibited RSV infection (Fig. S4B). Meanwhile, we found that SQLE knockdown of HEp-2 cells also showed a significant reduction in RSV infectivity (Fig. S4C). We chose two rate-limiting enzyme inhibitors in the cholesterol similar to MβCD. Treatment with simvastatin, a competitive inhibitor of the activity of the HMGCR, and terbinafine, a noncompetitive SQLE inhibitor, can effectively reduce the levels of cholesterol and

lipid rafts (Fig. S5). Consistently, the inhibitory effect of both drugs was observed only in the early stages of infection (0–6 hours post-infection [h.p.i.]), affecting viral entry (Fig. 5A). Dose-dependent inhibition was observed, with $EC_{50}$ values of 1.9 mM for MβCD, 2.0 µM for simvastatin, and 53.3 µM for terbinafine (Fig. 5B and C). CCK8 assay confirmed no significant cytotoxicity at these concentrations (Fig. 5C). These results indicated that cholesterol depletion inhibits RSV infectivity.

## Combined treatment with AK0529 and simvastatin mitigates RSV infection both *in vitro* and *in vivo*

We next analyzed the effect of combining the entry inhibitor simvastatin with the fusion inhibitor AK0529 against RSV (Fig. 6A and B). Both AK0529 and simvastatin inhibited RSV A2 and RSV B18537 replication in a dose-dependent manner, with the A2 virus strain exhibiting greater sensitivity than B18537 in HEp-2 cells. At lower concentrations, combination therapies with varying doses of the two drugs showed enhanced antiviral inhibition (Fig. 6A and B).

We assessed the antiviral activity of AK0529 at a concentration of 2 µM simvastatin and found that combination therapy significantly increased its antiviral potency. The $EC_{50}$ of AK0529 against the A2 virus decreased from 1.4 nM to 0.6 nM, while the $EC_{50}$ against the B virus decreased 10-fold (from 1.0 nM to 0.1 nM; Fig. 6C). The findings demonstrated that the combination of AK0529 and simvastatin led to an enhancement in antiviral potency.

To further evaluate whether AK0529 and simvastatin exhibit synergistic activity against RSV, the Zero Interaction Potency (ZIP) model was employed for further analysis. Checkerboard drug combination assays were conducted using AK0529 and simvastatin in HEp-2 cells. As shown in Fig. 6D and E, these results were obtained through analysis using SynergyFinder 3.0. The ZIP synergy score was calculated based on concentration gradients and inhibition indices (46). Drug combinations exhibiting ZIP synergy scores exceeding 5.0 or below −5.0 were classified as synergistic and antagonistic interactions, respectively, with scores falling within the range of −5.0 to 5.0 indicative of additive effects. The findings demonstrated that under the specified *in vitro* conditions, the combination of AK0529 and simvastatin significantly enhanced antiviral efficacy (ZIP synergy score for RSV A2 = 1.54 and B18537 = 3.24).

We then assessed the antiviral efficacy of a low-dose combination of AK0529 and simvastatin in a BALB/c mouse model infected with RSV A2. Mice received intranasal RSV ($5 \times 10^5$ PFU) and were treated orally for 4 days with AK0529 (12.5, 25, or 50 mpk), simvastatin (50 mpk), or their combinations (12.5 mpk or 25 mpk each; Fig. 7A). Monotherapy reduced lung viral titers significantly: AK0529 by 17.6-fold at 12.5 mpk ($P < 0.01$), 30.4-fold at 25 mpk ($P < 0.001$), and 71.7-fold ($P < 0.001$) at 50 mpk; simvastatin by 7.9-fold ($P < 0.05$; Fig. 7B). The combination treatments further enhanced this effect, reducing viral loads below detectable levels. Histological analysis revealed that while control mice exhibited severe lung abnormalities and immune cell infiltration, those treated with the drug combination showed minimal inflammation and intact alveolar structures (Fig. 7C). Pathology scores corroborated these findings, with the combination-treated group displaying the lowest severity of peribronchiolitis and alveolitis (Fig. 7D). These data confirm that RSV displays an enhanced block of infection upon the combined use of AK0529 and simvastatin *in vitro* and *in vivo*.

## DISCUSSION

RSV, an enveloped virus, utilizes multiple entry pathways to enter the cells for replication. Studying these entry pathways, especially those exploited by strains that are currently epidemic, can reveal universal mechanisms employed by RSV. Identifying such mechanisms is crucial for the development of effective antiviral drugs and strategies.

Most viruses utilize endocytosis to enter cells because this method delays immune responses and provides a mechanism for translocating across the plasma membrane (47). In this study, we describe the entry pathways utilized by RSV strains A2, B18537, and

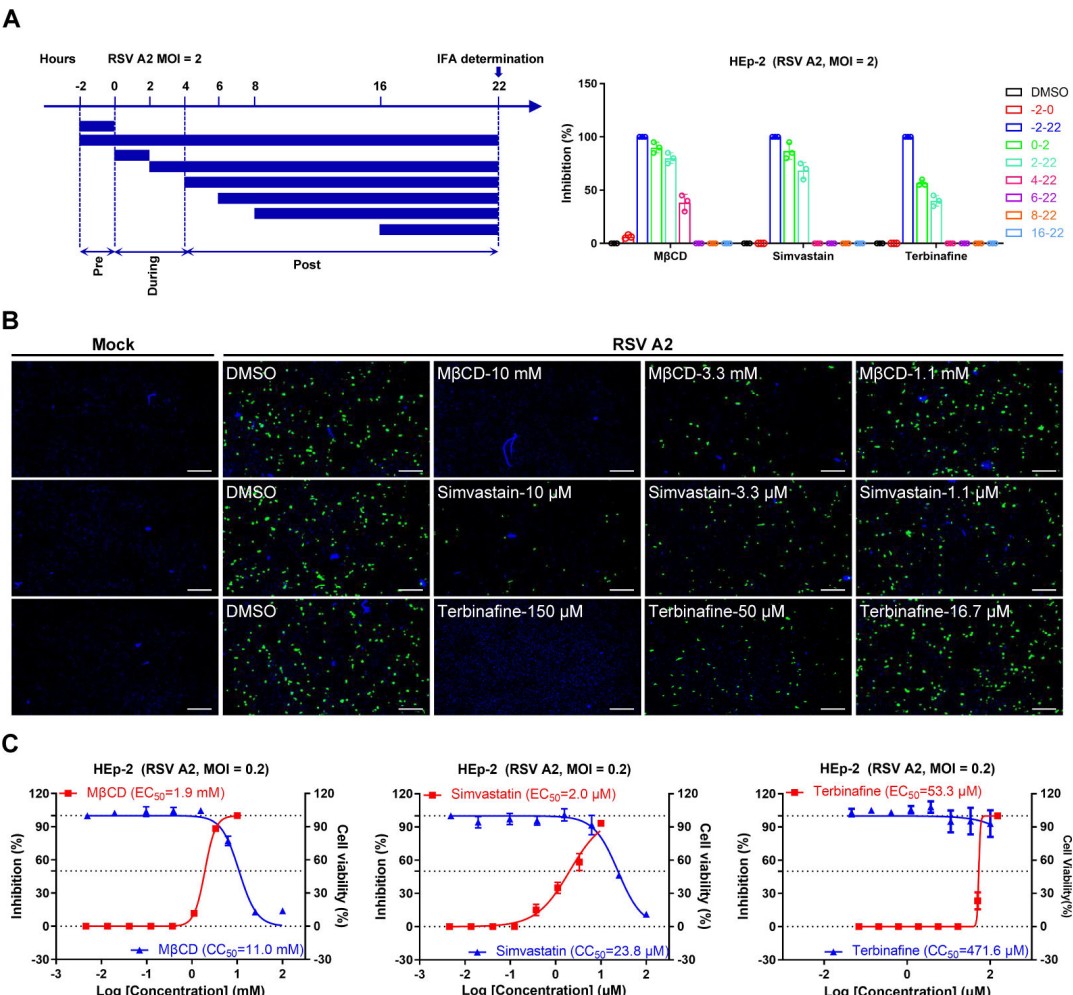

**FIG 5** Downregulation of cholesterol levels inhibits RSV entry and infection. (A) Schematic summary of the time-of-addition experiment. HEp-2 cells were infected with RSV A2 (MOI = 2) and treated with 5 mM MβCD, 5 μM simvastatin, or 100 μM terbinafine before (−2 to 0 hours), during (0–2 hours), and after (2, 4, 6, 8, and 16 hours) infection. DMSO (0.03%) served as the negative control. At 22 h.p.i., RSV infection rates were quantified by FFA to approximately capture a single cycle of infection. (B) Drug inhibition of viral infection assay. HEp-2 cells were pretreated with MβCD (1.1, 3.3, and 10 mM), simvastatin (1.1, 3.3, and 10 μM), or terbinafine (16.7, 50, and 150 μM) at the indicated concentration for 1 hour preceding the RSV exposure. The inhibitors were present in the media throughout the experiment. Cells were fixed after 24 hours, and RSV particles were detected by FFA. Scale bar = 300 μm. (C) Antiviral activity of MβCD, simvastatin, and terbinafine against RSV A2. The cytotoxicity of these compounds against HEp-2 cells was assessed by cell viability analysis. Data are representative of three independent experiments and are presented as mean ± SD.

the current epidemic ON1 strain into HEp-2, A549, and primary HBEC cells. We found that the cholesterol-rich lipid raft-mediated endocytosis pathway was the common pathway exploited by various RSV genotypes. Additionally, we demonstrated the recruitment of cellular receptors to lipid rafts, facilitating the binding of viral fusion glycoproteins and inducing actin rearrangement as well as endosome formation involving cholesterol-rich lipid rafts. Reducing membrane cholesterol content with MβCD, simvastatin, or terbinafine effectively inhibits RSV entry. Furthermore, combining simvastatin with AK0529, an inhibitor targeting viral fusion protein, resulted in an enhanced effect, enhancing therapeutic outcomes at lower drug concentrations.

It has been reported that RSV enters different cells through various endocytic pathways (18–22). RSV type A strains invade HEp-2 and HBEC cells using cholesterol-rich lipid rafts and actin-associated endocytosis (18, 19, 22), consistent with our findings. Caveolae-mediated endocytosis is only observed in A549 cells with RSV infection (20). Additionally, macropinocytosis has been observed in various cells with different

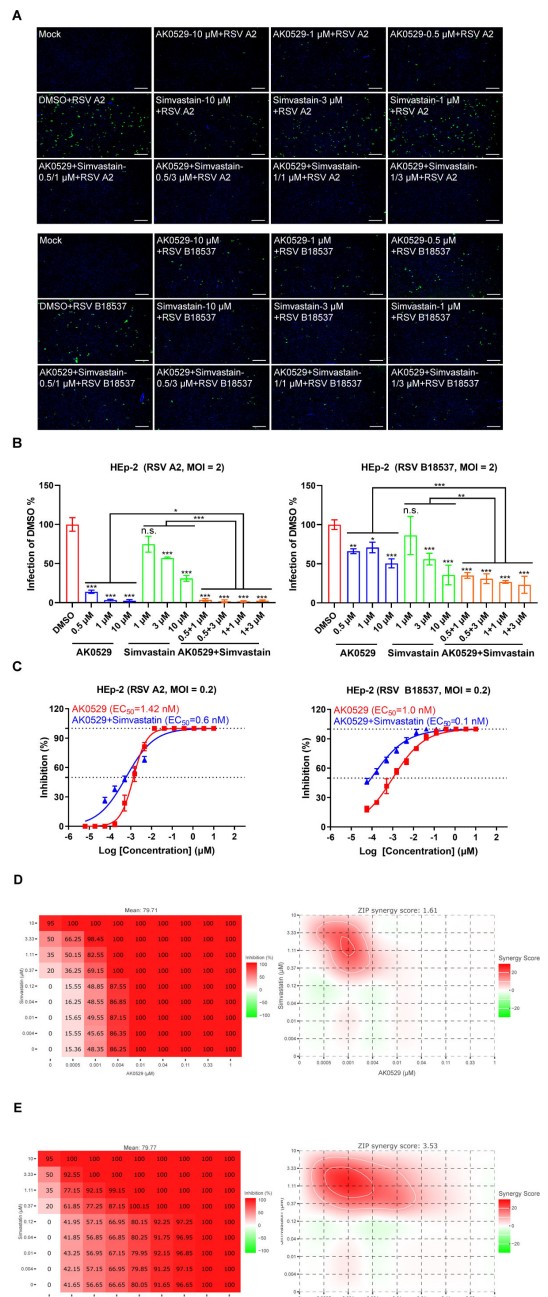

**FIG 6** *In vitro* assessment of the antiviral activity of AK0529 and simvastatin alone and/or in combination. (A) HEp-2 cells were preincubated for 1 hour at 37°C with specific inhibitors before viral exposure. Control received DMSO only. Cells were then infected with RSV A2 or B18537 (MOI = 2) in the presence of the inhibitor. After 2 hours, the medium with inhibitors was replaced, and RSV infections were measured by FFA 24 hours later using a fluorescence microscope. Scale bar = 300 μm. (B) The inhibition rate of the drug or drug combination was calculated by comparing the relative fluorescence intensity to that of the DMSO control group. Data are representative of three independent experiments and are presented as mean ± SD. (C) Evaluate the efficacy of AK0529 against RSV A2 and B18537 viruses in the presence of simvastatin (2 μM). (D and E) AK0529 synergizes with simvastatin to suppress RSV A2 or B18537 infection in HEp-2 cells. Cells were pretreated with AK0529 and simvastatin for 1 hour before infection at an MOI of 0.2. The medium was replaced at 2 h.p.i. After 72 hours, cytopathic effect (CPE) was assessed microscopically, and antiviral efficacy was calculated. The left panel shows infection inhibition, while the right panel presents drug synergy analysis using the Zero Interaction Potency (ZIP) model. Statistical differences were

Fig 6 (Continued)

determined by one-way analysis of variance (ANOVA) in B. *$P < 0.05$, **$P < 0.01$, and ***$P < 0.001$; n.s., not significant.

genotypes of virus infection, which is consistent with prior reports (21); however, the observed effect was relatively weak. It is plausible that the pathway inhibited by EIPA may not represent the predominant pathway by which RSV enters host cells. We found that B18537 enters host cells utilizing the cholesterol-rich lipid raft-mediated and actin-mediated endocytosis pathway, and both B18537 and the currently epidemic ON1 strains can employ the microtubule-mediated endocytosis pathway. These differences in cellular entry pathways among various RSV strains may provide new insights for clinical drug use.

Lipid rafts are specialized domains within the cell membrane characterized by rapid lateral mobility within the lipid bilayer, serving as platforms for protein attachment and internalization during signal transduction processes (23). Several enveloped viruses, including MARV (27), IAV (28), and SARS-CoV-2 (48), have been reported to utilize lipid rafts and actin-associated endocytosis pathways to enter cells. Lipid rafts create an environment conducive to efficient protein-protein interactions crucial for viral infectivity (49). RSV receptors NCL and IGF1R are recruited to lipid rafts, binding viral fusion proteins and enabling endocytosis. Actin is a scaffold for the assembly of the endocytic machinery, necessary for efficient viral invasion of the plasma membrane (42). RSV infection induces actin rearrangement, providing the driving force for membrane invagination and endosome formation involving the cholesterol-rich lipid rafts. After this initial entry, viruses are typically transported to the early endosomal compartment for sorting, followed by entry into late endosomes or fusion with early endosomes (44). In line with this scheme, RSV particles, APPL1, EEA1, Rab7, and STX6 significantly colocalized in endosomal components.

Cholesterol is a fundamental constituent of lipid rafts and plays crucial structural and signaling functions (24). Cholesterol accumulation in macrophages and other immune cells promotes inflammatory responses by augmenting Toll-like receptor (TLR) signaling and inflammasome activation (50). TLR-dependent signaling is important for the activation of the early inflammatory response to RSV, and aberrant TLR signaling exacerbates RSV-induced disease (51). Recently, it was suggested that RSV hijacks cholesterol or autophagy pathways to facilitate its optimal replication (32). Our works demonstrate that reducing cellular cholesterol levels effectively inhibited RSV entry and infection. Thus, cholesterol depletion may inhibit both viral entry and replication processes, as well as inflammatory responses. Therefore, cholesterol depletion may function through multiple mechanisms to exert an anti-RSV effect.

Therapeutic approaches utilizing a combination of drugs may enhance antiviral activity and improve clinical outcomes by allowing dose reduction, thereby decreasing dose-dependent toxicity and side effects (52). Combination therapy has shown remarkable success in the treatment of HIV (53), IAV (52), and SARS-CoV-2 (54). The combination of monoclonal antibodies (mAb) 130-6D and 131-2G against RSV G glycoprotein has also been reported to reduce lung inflammation in BALB/c mice (55). The safety and efficacy of small-molecule RSV fusion protein inhibitors (such as AK0529, GS-5806, and RV521) remain the subject of ongoing clinical trials, especially when administered orally at high doses (56–58). Here, we demonstrated that the simultaneous blocking of RSV fusion proteins (AK0529) and the inhibition of cholesterol synthesis (simvastatin) have an enhanced antiviral effect that could be exploited to reduce the necessary doses of these drugs both *in vitro* and *in vivo*.

In summary, we propose a model for RSV entry into cells. In this sequence, RSV infection induces the transport of proteins that act as RSV receptors to cholesterol-rich lipid rafts. The consequent binding between receptors and viral proteins triggers actin rearrangement, promoting endosome formation. This scheme explains why the use of simvastatin targeting the HMGCR enzyme in the host cholesterol biosynthesis pathway

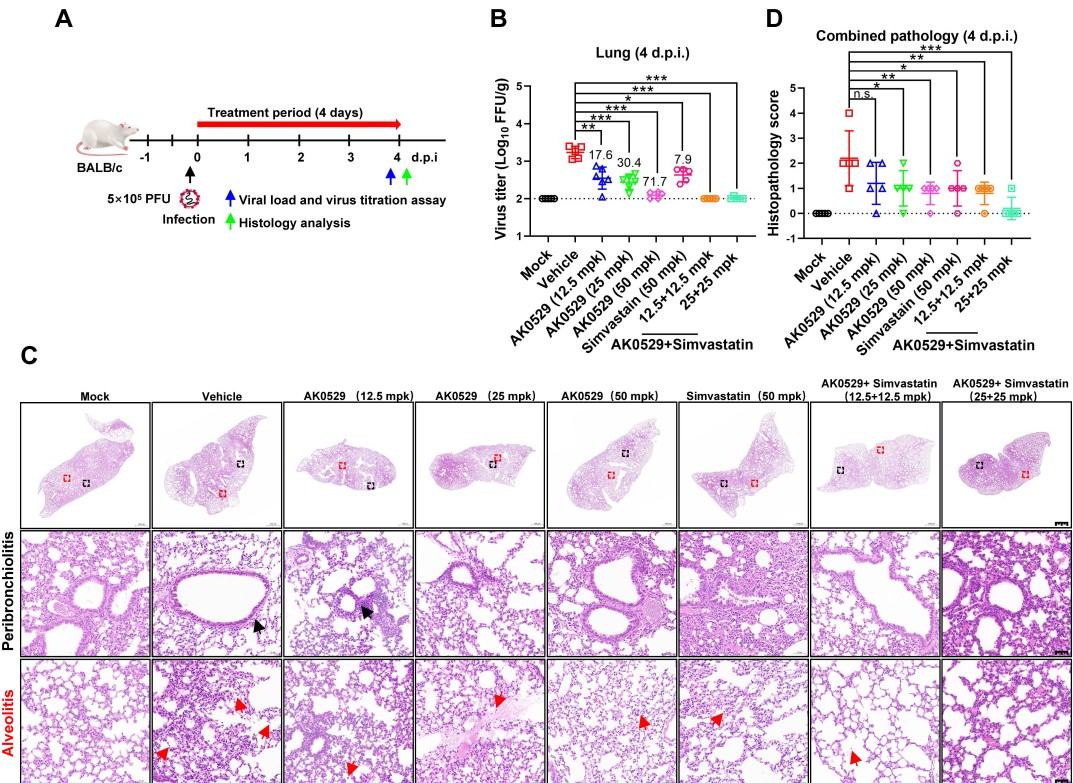

**FIG 7** *In vivo* assessment of the antiviral activity of AK0529 and simvastatin alone and/or in combination. BALB/c mice (*n* = 5 mice/group) received 12.5, 25, or 50 mpk of AK0529, 50 mpk of simvastatin, a combination of 12.5 mpk AK0529 + 12.5 mpk simvastatin, or 25 mpk AK0529 + 25 mpk simvastatin, or vehicle solution twice daily, only for the control group for 4 days post-infection (d.p.i.). On the day of infection, mice were inoculated intranasally with $5 \times 10^5$ PFU RSV A2 or media alone as control. (A) Schematic diagram summarizing the schedule of infection and treatment. (B) Mice infected with RSV A2 were killed at 4 d.p.i. for detection of infectious viral titers using FFA. (C) Histology of virus-induced pathology in the lungs of infected mice. Two parameters of pulmonary inflammation were evaluated: (i) peribronchiolitis (black arrows showing inflammatory cell infiltration around the bronchioles); (ii) alveolitis (red arrows: inflammatory cells within the alveolar spaces). Scale bar = 1,000 μm or 50  μm. Samples were processed from each of the five mice in all the treatment groups, and five sections were analyzed from each processed lung sample. (D) Quantitative pathology scores of lung tissues. Blind scores by two pathologists were given based on the severity of peribronchiolitis and alveolitis on a scale of 0–4 (0 is a normal healthy lung, and 4 is severe confluent areas of pathology). Each dot represents one mouse at the indicated time point. Data are representative of at least two experiments. Error bars show mean ± SD. Statistical differences were determined by one-way analysis of variance (ANOVA) in panel D. *$P <$ 0.05, **$P <$ 0.01, and ***$P <$ 0.001; n.s., not significant.

inhibits RSV entry and infection. Combining this benefit with the administration of the fusion inhibitor AK0529 results in enhanced antiviral effect *in vitro* and *in vivo*. Overall, our findings provide additional insight into the mechanisms involved in viral-host interaction during virus entry, providing a novel approach for the treatment of RSV infection.

## MATERIALS AND METHODS

### Cells and viruses

The human laryngeal epidermoid carcinoma (HEp-2) and human non-small cell lung cancer (A549) cell lines were cultured in Dulbecco's Modified Eagle Medium (DMEM, Hyclone), while primary HBECs were cultured in the specific medium CM-H009 (Procell). Human embryonic kidney 293T (HEK293T) cells were maintained in DMEM (Gibco). All media were supplemented with 10% fetal bovine serum (Gibco) and 1% Pen Strep (Gibco). All cells were tested for mycoplasma and were kept at 37°C in a humidified atmosphere of 5% $CO_2$.

RSV A2 (Genebank: KT992094), B18537 (Genebank: MG813995), and ON1 (GenBank: MW582528) strains were propagated in HEp-2 cells using established protocols (59). Briefly, 3 days post-infection, the supernatant was collected and centrifuged at 1,000 × $g$ for 10 min at 4°C. Viral titer was determined in HEp-2 cells by $TCID_{50}$/mL and fluorescence focus assay (FFA) in confluent cells in 96‐well microtiter plates. The titrated virus suspension was stored at −80°C.

## Plasmids

Gene fragments encoding the codon-optimized F/G/SH proteins of different RSV strains, dynamin-2, NCL-flag, and IGF1R-flag were cloned individually into a pcDNA3.1(−) vector (Beijing Tsingke Biotech Co., Ltd.). The caveolin-1, caveolin-1-DN, and dynamin-2-DN were constructed for use in this study. The pCMV-VSV-G and pLVX-IRES-Puro were purchased from MiaoLingBio and Beijing Tsingke Biotech Co., Ltd., respectively. The pMD2.G was preserved in our lab. The psPAX2 was kindly provided by Prof. Xiancai Ma.

## Antibodies, inhibitors, and reagents

The RSV-FITC antibody (GTX36375) was purchased from GeneTex. The RSV [2F7] antibody (ab43812) was purchased from Abcam. The nucleolin polyclonal antibody (10556-1-AP), IGF1R beta chain polyclonal antibody (20254-1-AP), APPL1 monoclonal antibody (68195-1-Ig), EEA1 monoclonal antibody (68065-1-Ig), DNM2 monoclonal antibody (68209-1-Ig), GAPDH monoclonal antibody (60004-1-Ig), and Beta Actin Monoclonal antibody (66009-1-Ig) were all purchased from Proteintech. The Flotillin-1 (D2V7J) rabbit mAb (#18,934T), caveolin-1 (D46G3) rabbit mAb (#3267), EEA1 (C45B10) rabbit mAb (#3288), Rab7 (D95F2) rabbit mAb (#9367), Rab7 (E9O7E) mouse mAb (#95746), and syntaxin 6 (C34B2) rabbit mAb (#2869) were purchased from Cell Signaling Technology. The Alexa Fluor 568-labeled donkey anti-rabbit IgG (A10042), Alexa Fluor 568-labeled donkey anti-mouse IgG (A10037), and Alexa Fluor 633-labeled goat anti-mouse IgG (A21052) secondary antibodies were from Thermo Fisher Scientific.

The chemical inhibitors used were umifenovir hydrochloride (Arbidol, HY-14904A, MCE), ziresovir (AK0529, HY-109142, MCE), presatovir (GS-5806, HY-16727, MCE), $NH_4Cl$ (A9434, Sigma), filipin complex (HY-N6716, MCE), CPZ (S5749, Selleck), EIPA (HY-101840, MCE), MβCD (HY-101461, MCE), cytochalasin D (CytoD, HY-N6682, MCE), Noc (HY-13520, MCE), palivizumab (HY-P9944, MCE), simvastatin (HY-17502, MCE), and terbinafine (HY-17395A, MCE).

DAPI (D3571), 1′-dioctadecyl-3,3,3′,3′-tetramethylindodicarbocyanine (DiD; D7757), Alexa Fluor-488/647-conjugated cholera toxin subunit B (Alexa 488/647-CTB, C34775/C34778), NBD-cholesterol (N1148), and the CellMask orange actin tracking stain (A57244) were all from Invitrogen.

## Focus formation assay

A total of $1.8 \times 10^4$ HEp-2 cells were seeded in 96-well plates 20 hours prior to infection. These cells were infected with RSV in medium, after 24 hours fixed in 4% formaldehyde, permeabilized with 0.3% Triton X-100, and incubated with 5% BSA (B824162, Macklin) for 30 min. These pretreated cells were then stained with RSV-FITC antibody (1:100). RSV-FITC was observed using the PerkinElmer Operetta CLS, and fluorescence intensity was quantified using ImageJ.

## Evaluation of the *in vitro* antiviral activity

For Fig. 1A; Fig. S1, to test the effect of inhibitors on RSV invasion, cells were preincubated with a medium containing the specific inhibitors at 37°C for 1 hour before viral exposure. Subsequently, cells were infected with RSV A2, B18537, or ON1 (MOI = 2) in the medium that also contained the specific inhibitor used in a given experiment. The tissue culture medium without inhibitor was replaced after 2 hours of incubation, and 24 h.p.i., the infection rate was quantified using an FFA.

For Fig. 5B and C, HEp-2 cells ($1.8 \times 10^4$) were seeded in 96-well plates for 20 hours. Cells were preincubated with a medium containing the MβCD, simvastatin, or terbinafine at 37°C for 1 hour before viral exposure. Subsequently, cells were infected with RSV A2 (MOI = 0.2) in the medium that also contained the specific inhibitor used in a given experiment. The virus inhibition rate was quantified via FFA at 24 h.p.i. The antiviral activity of the compounds was expressed as $EC_{50}$, which meant the drug concentration required to achieve 50% of the inhibition rate, and was calculated using nonlinear regression and GraphPad Prism 8.0 software.

## Virus binding and internalization assay

HEp-2 cells ($1.5 \times 10^5$ per well) were seeded on 24-well plates for 20 hours. For the binding assay, cells were preincubated with a medium containing MβCD (5 mM) at 37°C for 1 hour before viral. Subsequently, the cells were infected with RSV A2, B18537, or ON1 (MOI = 10) on ice for 1 hour, respectively. Palivizumab (10 µg/mL) was used as a control. The cells were lysed in TRIzol for RNA extraction after five cycles of washing. For the internalization assay, after five cycles of washing, the cells were added to the medium supplemented with MβCD (5 mM) or palivizumab (10 µg/mL) and then transferred to a 37°C incubator for 1 hour. Thereafter, freeze cells on ice and then treat with proteinase K (500 ng/mL) on ice for 1 hour. After five additional washes, the cells were collected for RNA extraction. Afterward, the viral RNA level was determined using quantitative real-time PCR (qRT-PCR). Data are representative of three independent experiments.

## Plasmid construction

The *Xba*I and *Bam*HI restriction enzyme cleavage site containing primers (primer-F: 5′-T GCTCTAGAGCCACCATGAAGTCTGGCT-3′, primer-R: 5′-CGCGGATCCTCAATGGTGATGGTGAT GG-3′) were used to amplify the full-length IGF1R cDNA. After enzymatic cleavage using the restriction enzyme sites, the product was cloned into the pLVX-IRES-puro vector. Using a similar strategy, the full-length NCL cDNA was amplified, cleaved, and ligated into the pIRES-puro vector using *Not*I and *Bam*HI restriction sites.

Caveolin-1 and dynamin were amplified from HEp-2 cDNA using primers containing the *Xba*I and *Kpn*I restriction sites (caveolin-1-primer-F: 5′-GCTCTAGAGCCACCATGTCT GGGGGGCAAATACGTAGA-3′, caveolin-1-primer-R: 5′-GGGGTACCTTATATTTCTTTCTGCAAGT TGATGC-3′), and the cleaved product was inserted into the pcDNA3.1(−) vector. DN mutant sequences were constructed based on previous reports (60, 61). To construct mutations/deletions corresponding to functional domains, the full-length cDNA was reamplified following the following primer sets: caveolin-1 DN (deletion of nucleotides 82–178; caveolin-1-DN-primer-F: 5′-GAAGGGACACACAGTTTTTAAGGTACCAAGCTTAA G-3′, caveolin-1-DN-primer-R: 5′-CTTAAGCTTGGTACCTTAAAAACTGTGTGTCCCTTC-3′) and dynamin-2 DN (K44A; dynamin-2-DN-primer-F: 5′-TGGGCGGCCAGAGCGCCGGCGCGAGC TCGGTGCTGGAGAACTT-3′, dynamin-2-DN-primer-R: 5′-AAGTTCTCCAGCACCGAGCTCGCG CCGGCGCTCTGGCCGCCCA-3′).

## Establishment of stable cell lines

HEK293T cells were seeded into six-well plates. After 1 day of incubation, the cells were transfected with 0.75 µg pspAX2, 0.25 µg pMD2.G, and 1 µg pLVX-NCL or pLVX-IGF1R construct, using Lipofectamine 2000 (11668019, Invitrogen). A 1.5-fold volume of a Lipofectamine 2000 working solution was mixed with the indicated amounts of DNA diluted in Opti-MEM medium. Supernatants were harvested after 72 hours by centrifugation at $4{,}000 \times g$ for 10 min at 4°C, and cellular debris was cleared by passaging the supernatant through a 0.45 µm pore size filter. The supernatant was added to polybrene at a final concentration of 10 µg/mL (sc-134220, Santa Cruz Biotechnology), and the mixture was used to infect HEp-2 cells. After 24 hours, the culture medium was replaced with a fresh medium, and after 48 hours of additional incubation, a medium containing 2 µg/mL puromycin was added for 6 days to screen for positive colonies.

## RNA interference

HEp-2 cells grown in sixwell plates were transfected with siRNA (20 nM) using Lipofectamine RNAiMAX reagent (13778150, Invitrogen) according to the manufacturer's instruction. Non-silencing siRNA with a scrambled sequence was used as a negative control (siNC). The NCL, IGF1R, HMGCR, and SQLE-specific siRNA sequences (sense/antisense, 5′−3′) were NCL#1, CGGCUUUCAAUCUCUUUGUTT/ACAAAGAGAUU-GAAAGCCGTT; NCL#2,CGGCUUUCAAUCUCUUUGUTT/UAACUGUCUUCUUGGCAGGTT; NC L#3, GGCGAUCUAUUUCCCUGUATT/UACAGGGAAAUAGAUCGCCTT; IGF1R#1, GGAGAGAA CUGUCAUUUCUTT/GGAGAGAACUGUCAUUUCUTT; IGF1R#2, GCGGUGUCCAAUAACUACA TT/UGUAGUUAUUGGACACCGCTT; IGF1R#3, GCAUACCUCAACGCCAAUATT/UAUUGGCGU UGAGGUAUGCTT; HMGCR#1, GCAUAGCCAUCCUGUAUAUTT/AUAUACAGGAUGGCUAUGC TT; HMGCR#2, GUCCCAGACAAUUGUUGUAUT/UACAACAAUUGUCUGGGACTT; HMGCR#3, CUGGCUCGAAACAUCUGAATT/UUCAGAUGUUUCGAGCCAGTT; SQLE#1, GGUGUUGUGUU ACAGUUAUTT/AUAACUGUAACACAACACCTT; SQLE#2, GCUCAGGCUCUUUAUGAAUTT/A UUCAUAAAGAGCCUGAGCTT; SQLE#3, GCCUGUACAUCAACAUCUUTT/AAGAUGUUGAU-GUACAGGCTT. To maintain efficient silencing during the entire experiment, cells were transfected with the same siRNA again after the first 24 hours. RSV infection was performed 24 hours after the second transfection.

## siRNA knockdown and plasmid rescue

HEp-2 cells, at a density of $1 \times 10^5$ cells per well, were cultured in 24-well plates for a duration of 20 hours. Subsequently, the cells underwent transfection with siRNA targeting either NCL or IGF1R at a concentration of 40 nM to induce gene knockdown. A subset of these cells was further subjected to transfection with plasmids expressing NCL or IGF1R (500 ng/well) to facilitate functional rescue. After these interventions, selected samples from both groups were pretreated with 2 mM MβCD for 1 hour, whereas control samples received a vehicle treatment consisting of 0.1% DMSO. Following these treatments, all cells were infected with RSV A2 or B18537 (MOI = 2) for 2 hours in the medium that also contained 2 mM MβCD. The tissue culture medium without inhibitor was replaced after 2 hours of incubation, and at 24 h.p.i., the infection rate was quantified using an FFA. This involved fixing the cells, immunostaining with RSV-FITC antibody, and analyzing the relative fluorescence intensity, which was normalized to the DMSO-treated siNC control group, set as 100% infection. The data are presented as mean ± SD from three independent experimental replicates.

## RNA extraction and qRT-PCR

Total RNA was extracted from RSV-infected HEp-2 cells using Trizol reagent. The abundance of specific gene transcripts was analyzed by one-step real-time qRT-PCR with specific primers and the HiScript II One Step qRT-PCR SYBR Green Kit (Q221-01, Vazyme) on an Applied Biosystems QuantStudio 6 Flex instrument. The primer sequences (F/R, 5′−3′) to amplify the targets F gene were CGAGCCAGAAGAGAACTACCA/CCTTCTAGGTG CAGGACCTTA; for β-actin: CTCGACACCAGGGCGTTATG/CCACTCCATGCTCGATAGGAT; for NCL: TTGAGGGCAGAGCAATCAGG/AGAGTTTTGGATGGCTGGCT; for IGF1R: CCTGCACAAC TCCATCTTCGTG/CGGTGATGTTGTAGGTGTCTGC; for HMGCR: GACCAACCTACTACCTCAGCA AGC/CAGCCATTACGGTCCCACACAC; for SQLE: TGACAATTCTCATCTGAGGTCCA/TGACAATT CTCATCTGAGGTCCA; for β-actin Mouse: AGTGTGACGTTGACATCCGT/GCAGCTCAGTAACA GTCCGC. Relative quantification was based on housekeeping gene control levels.

## Western blotting

Whole-cell lysates were prepared using a lysis buffer containing 50 mM Tris-base (pH 7.5), 1 mM EGTA, 1 mM EDTA, 1% Triton X-100, 150 mM NaCl, 100 µM phenylmethyl-sulfonyl fluoride, and protease inhibitors (Roche) for 30 min in ice (62). Cell lysates were centrifuged at $10,000 \times g$ for 10 min at 4°C. The recovered supernatants were

denatured at 95°C for 10 min. The samples were resolved by SDS-PAGE and transferred to nitrocellulose. Membranes were blocked with TBST (pH 7.4, containing 0.1% Tween‑20) containing 5% skimmed milk for 1 hour at room temperature. This was followed by incubation with the Flotillin-1 monoclonal antibody (1:1,000), NCL polyclonal antibody (1:1,000), IGF1R beta chain polyclonal antibody (1:1,000), and GAPDH monoclonal antibody (1:1,000) as primary antibodies, overnight at 4°C. After washing, the membranes were incubated for 1 hour at room temperature with horseradish peroxidase‑conjugated secondary antibodies and imaged using the FluorChem HD2 system (Alpha Innotech).

## Plasma membrane-derived lipid rafts isolation

Cells were infected with RSV A2 (MOI = 5) at 37°C for 60 min. Approximately $4 \times 10^7$ cells were harvested, washed with ice-cold PBS, and centrifuged at $500 \times g$ for 10 min at room temperature. Lipid rafts were extracted using the Minute plasma membrane-derived lipid raft isolation kit (Invent Biotechnologies, USA). The obtained cell pellets were incubated on ice with Buffer A (containing phosphatase and protease inhibitors), followed by centrifugation at $500 \times g$ for 10 min at room temperature. Then, the pellets were treated with the nonionic detergent-containing buffer, and the aqueous phase was removed. The resulting highly enriched plasma membrane fractions were used as the lipid rafts. Flotillin-1, NCL, IGF1R, and GAPDH were separated by SDS-PAGE and detected by WB analysis.

## Receptor localization assays

A total of $2 \times 10^5$ HEp-2 or A549 cells were seeded onto glass dishes for 24 hours. Cells were infected with RSV A2 (MOI = 30) at 37°C for 60 min after transfection with the pcDNA3.1-NCL or IGF1R plasmids for 24 hours. The infected cells were fixed with 4% paraformaldehyde for 15 min and permeabilized in 0.3% Triton X-100 for 15 min at room temperature. The glass dish was blocked in 5% BSA in phosphate‑buffered saline for 1 hour. NCL expression was detected with NCL polyclonal antibody (1:200), and IGF1R was probed with IGF1R beta chain polyclonal antibody (1:200) and an Alexa 568‑labeled donkey anti‑rabbit IgG (1:500) as the secondary antibody. RSV was then labeled with either an RSV-FITC (1:100) or an unlabeled RSV [2F7] antibody (1:100) using an Alexa Fluor 633-linked goat anti-mouse IgG (1:500) as the secondary antibody. Lipid rafts and cholesterol were stained with Alexa 647-CTB (1:200) and NBD-cholesterol (15 µM) at 37°C for 30 min, respectively. Fluorescent images were acquired using the light scanning module of a NIKON A1 confocal microscope.

## Caveolin, dynamin, or clathrin-mediated endocytosis assays

A total of $2 \times 10^5$ HEp-2 cells were seeded in glass dishes for 24 hours. Cells were infected with RSV A2 (MOI = 30) at 37°C for 90 min after transfection with the following plasmids: pcDNA3.1-caveolin-1, pcDNA3.1-dynamin-2, pcDNA3.1-caveolin-1-DN, and pcDNA3.1-dynamin-2-DN for 24 hours. Samples were fixed and permeabilized as described above. Caveolin-1 was detected with a caveolin-1 rabbit mAb (1:200), DNM2 with a DNM2 Monoclonal antibody and an Alexa 568-labeled donkey anti‑rabbit IgG (1:500) or Alexa 568-labeled donkey anti-mouse IgG (1:500) as the secondary antibody. RSV, lipid rafts, and cholesterol were stained as described above. Fluorescent images were acquired using the light scanning module of a NIKON A1 confocal microscope.

## Actin polymerization assay

A total of $2 \times 10^5$ HEp-2 or A549 cells were seeded in a glass dish for 24 hours and infected with RSV A2 (MOI = 30) at 37°C for 90 min. The infected cells were fixed with 4% paraformaldehyde for 15 min and permeabilized in 0.3% Triton X-100 for 15 min at room temperature. The glass dish was blocked in 5% BSA in PBS for 1 hour. RSV particles were detected with an RSV-FITC (1:100) or unlabeled RSV [2F7] antibody (1:100) and an

Alexa Fluor 633-linked goat anti-mouse IgG (1:500) as the secondary antibody. Actin was stained using the CellMask orange actin tracking stain (1:1,000), while lipid rafts and cholesterol were stained as described above. Confocal microscopy was carried out using a NIKON A1 instrument.

## Internalization assay

A total of $2 \times 10^5$ HEp-2 or A549 cells were seeded in glass dishes for 24 hours and infected with RSV A2 (MOI = 30) at 37°C for 60 or 150 min. Samples were fixed and permeabilized as described above. APPL1, EEA1, Rab7, or STX6 expression was detected using APPL1, EEA1, Rab7, and STX6 mAbs (1:100) followed by an Alexa 568-labeled donkey or rabbit IgG (1:500) as the secondary antibody. The staining of RSV, lipid rafts, cholesterol, and image acquisition were carried out as described elsewhere.

## Cholesterol depletion and quantification

HEp-2 cells were treated with the inhibitors at the following concentrations: MβCD (2, 5, and 10 mM), simvastatin (1.1, 3.3, and 10 µM), and terbinafine (16.7, 50, and 150 µM) for 1 hour or 12 hours, respectively. Cells were fixed (4% PFA, 15 min, room temperature), and fluorescent images were acquired using a NIKON A1 confocal microscope. Fluorescence intensity was quantified using the ImageJ software. Relative quantification was performed against a control group, and statistical plots were generated with GraphPad Prism 8.0.

## Live-cell microscopy

RSV suspension was labeled with DiD for 90 min at room temperature, at a final DiD concentration of 45 µM, and the free label was separated using 0.45 µm pore size filters. Before infection with RSV-DiD, lipid rafts and cholesterol were stained with Alexa 488-CTB (1:200) and NBD-cholesterol (15 µM) at 37°C for 30 min.

## Dual-drug antiviral regimen design

Checkerboard assays of two drug combinations were performed as described (46). Two drugs were tested in a $9 \times 9$ matrix with seven concentrations of each drug, including a DMSO control. HEp-2 cells ($1.5 \times 10^4$ per well) were seeded in 96-well plates and infected with RSV A2 or B18537 (MOI = 0.2). Each drug combination was tested in duplicate. Antiviral activity was assessed at 72 hours post-infection by measuring the reduction in CPE. The Celigo software module "Confluence 1" was employed to determine the percentages of host cell confluence by analyzing the acquired bright-field images. The confluence percentages were calculated as the ratio of the cell-covered surface area to the total surface area within the well, as directly measured by the image cytometer (63, 64). Dose-response data from checkerboard assays were analyzed in SynergyFinder 3.0.

## Cytotoxicity assay

HEp-2 cells were seeded in 96-well plates and cultured in the presence of selected concentrations of the used inhibitors. After 72 hours of incubation, cell viability was assessed using the CCK-8 (C0039, Beyotime) kit according to the manufacturer's instructions.

## Time-of-drug addition assays

A time-of-addition assay was performed to determine the step in the viral life cycle inhibited by 5 mM MβCD, 5 µM simvastatin, and 100 µM terbinafine. HEp-2 cells were infected with RSV (MOI = 2) for 2 hours at 37°C, and MβCD, simvastatin, and terbinafine were added at the following time points: pre-infection (−2 to 0 hours), during infection (0 to 2 hours), and post-infection (2, 4, 6, 8, and 16 hours). DMSO (0.03%) was used as a

control. RSV infection rates were quantified by FFA 24 h.p.i. to approximately capture a single cycle of infection.

## *In vivo* efficacy

Thirty-five 8-week-old female BALB/c mice were purchased from GemPharmatech Co., Ltd (Nanjing, China) and housed in a specific pathogen-free environment under standard conditions. *In vivo* antiviral activity studies were approved by the Ethics Committee of Guangzhou National Laboratory Animals (GZLAB-AUCP- 2024-01-A4). On the day of infection, mice were anesthetized with isoflurane and infected intranasally with $5 \times 10^5$ PFU of RSV A2. Drug therapy with AK0529 (12.5, 25, or 50 mpk), simvastatin (50 mpk), AK0529 (12.5 mpk) + simvastatin (12.5 mpk), or AK0529 (25 mpk) + simvastatin (25 mpk) was initiated 2 hours after the initial infection. The used inhibitors/inhibitor combinations were administered orally, with the drugs dissolved in 200 µL 5% DMSO/20% hydroxypropyl-β-cyclodextrin. The control group was treated with the vehicle alone. After 4 days of treatment, the mice were euthanized, and lung tissue was sampled for virological and histopathological analyses. After weighing the right lung tissue, 1,000 µL PBS was added, and the sample was homogenized. A total of 100 µL of homogenized tissue was used to determine the virus titer by FFA. Briefly, the homogenized lung tissue was spun, and a fourfold dilution of the supernatant was used to infect tissue culture wells. The next day, cells were fixed and permeabilized, and the RSV particles were stained and quantified as described elsewhere. Viral titers were calculated according to the following formula: virus titer (FFU/g) = average number of fluorescent foci per well × dilution degree × volume index × volume × lung tissue weight index (converted to g). The left lung tissue was fixed in tissue fixative solution, embedded, sectioned, and stained with H&E to observe pathologic changes. Slides were scored blindly by two independent pathologists on a 0–4 severity scale (65).

## Statistical analysis

Data on figures represent mean ± SD. All data were analyzed with GraphPad Prism 8.0 software. Statistical comparison between different groups was performed by one-way analysis of variance and Student's *t*-test. *P*-values were calculated, and statistical significance was reported as highly significant with *$P < 0.05$, **$P < 0.01$, ***$P < 0.001$, and ****$P < 0.0001$; n.s., not significant.

## ACKNOWLEDGMENTS

We acknowledge the Advanced Bio-imaging Technology Platform of Guangzhou Laboratory for continuous and generous support.

This work was supported by the Natural Science Foundation of Guangdong Province (grant No. 2024A1515011589 to Q.Y.); the National Natural Science Foundation of China (grant No. 32470168 to Q.Y.); the Pearl River Talent Recruitment Program (grant No. 2019 No. 2019CX01Y422 to X.C.); the Guangzhou Laboratory (grant No. SRPG22-002 to X.C.; No. SRPG22-011 to Q.Y.); and the Basic and Applied Basic Research Projects of Guangzhou Basic Research Program (2023A04J0161 to Q.Y.).

Q.Y., X.C., and J.T. conceived the project. A.Z. and B.X. conducted RSV propagation assay. A.Z. and B.X. conducted the cell-based antiviral assays and cellular cytotoxicity assays. A.Z. carried out the live cell imaging and cellular immunofluorescence assays. A.Z., B.X., and J.L. performed in vivo antiviral studies. J.L. and A.Z. cloned and expressed plasmids. J.L. and Y.Z. helped with project management. Q.Y., X.C., J.T., A.Z., B.X., J.Z., F.W., and R.P. analyzed and discussed the data. The article was written by Q.Y., X.C., and A.Z. with input from all the authors.

## AUTHOR AFFILIATIONS

[1]State Key Laboratory of Respiratory Disease, Guangzhou Medical University, Guangzhou, Guangdong, China
[2]Guangzhou National Laboratory, Guangzhou, Guangdong, China
[3]Wuhan Institute of Virology, Chinese Academy of Sciences, Wuhan, China
[4]University of Chinese Academy of Sciences, Beijing, China

## AUTHOR ORCIDs

Jielin Tang ⓘ http://orcid.org/0000-0002-4398-2927
Qi Yang ⓘ http://orcid.org/0009-0009-1648-4189
Xinwen Chen ⓘ http://orcid.org/0000-0002-4052-8155

## FUNDING

| Funder | Grant(s) | Author(s) |
| --- | --- | --- |
| Natural Science Foundation of Guangdong Province | 2024A1515011589 | Qi Yang |
| National Natural Science Foundation of China | 32470168 | Qi Yang |
| Guangdong Provincial Pearl River Talents Program | 2019CX01Y422 | Xinwen Chen |
| Major Project of Guangzhou National Laboratory | SRPG22-002 | Xinwen Chen |
| Major Project of Guangzhou National Laboratory | SRPG22-011 | Qi Yang |
| Basic and Applied Basic Research Project of Guangzhou Basic Research Program | 2023A04J0161 | Qi Yang |

## DATA AVAILABILITY

The complete sequence of RSV A2 (GenBank: KT992094), RSV B18537 (GenBank: MG813995), and RSV ON1 (GenBank: MW582528) is available on GenBank. Additional data are provided in Supplementary information. Source data are provided with this paper.

## ADDITIONAL FILES

The following material is available online.

### Supplemental Material

**Supplemental figures and videos legends (Spectrum01192-25-s0001.pdf).** Fig. S1 to S5 and Legends for video S1 to S4.
**Video S1 (Spectrum01192-25-s0002.mp4).** HEp-2 cells lipid rafts colocalize with DiD-labeled RSV particles during viral entry.
**Video S2 (Spectrum01192-25-s0003.mp4).** HEp-2 cells cholesterol colocalizes with DiD-labeled RSV particles during viral entry.
**Video S3 (Spectrum01192-25-s0004.mp4).** A549 cells lipid rafts colocalize with DiD-labeled RSV particles during viral entry.
**Video S4 (Spectrum01192-25-s0005.mp4).** A549 cells cholesterol colocalizes with DiD-labeled RSV particles during viral entry.

### Open Peer Review

**PEER REVIEW HISTORY (review-history.pdf).** An accounting of the reviewer comments and feedback.

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
