## [Reviewer comments · Microbiology Spectrum]

Microbiology Spectrum

Cholesterol-Rich Lipid Rafts Mediate Endocytosis as a Common Pathway for Respiratory Syncytial Virus Entry into Different Host Cells

Anqi Zhou, Bao Xue, Jiayi Zhong, Junjun Liu, Ran Peng, Fan Wang, Yuan Zhou, Jielin Tang, Qi Yang, and Xinwen Chen

Corresponding Author(s): Xinwen Chen, Guangzhou Laboratory

Review Timeline:

Submission Date:	April 22, 2025
Editorial Decision:	June 1, 2025
Revision Received:	June 5, 2025
Editorial Decision:	June 16, 2025
Revision Received:	June 16, 2025
Accepted:	June 18, 2025

Editor: Leiliang Zhang

Reviewer(s): Disclosure of reviewer identity is with reference to reviewer comments included in decision letter(s). The following individuals involved in review of your submission have agreed to reveal their identity: Jinsheng He (Reviewer #2)

Transaction Report:

DOI: <https://doi.org/10.1128/spectrum.01192-25>

Re: Spectrum01192-25 (**Cholesterol-Rich Lipid Rafts Mediate Endocytosis as a Common Pathway for Respiratory Syncytial Virus Entry into Different Host Cells**)

Dear Prof. Xinwen Chen:

Thank you for the privilege of reviewing your work. Below you will find my comments, instructions from the Spectrum editorial office, and the reviewer comments.

Revision Guidelines

Sincerely,
Leiliang Zhang
Editor
Microbiology Spectrum

Reviewer #2 (Comments for the Author):

Major concerns:

Page 6, lines 93-94, "RSV infection causes 94 lung inflammation and asthma by cholesterol-dependent pathways (30-32)." no opinions or data presented in these refs demonstrated the lung inflammation and asthma caused by cholesterol-dependent pathways following RSV infection. For example, the opposite opinions, both enhancement and suppression of inflammation

response from this pathway, were introduced in ref 30, no information mentioned on the cholesterol-dependent pathways following RSV infection in ref 31, and no direct evidence support this overexpression of NLRP3 inflammasome (critical in RSV-induced asthma) via this pathway following the elevated expression of orosomucoid 1-like protein 3 (ORMDL3) after RSV infection in ref 32. Please explain it.

Page 12, lines 234-235, if receptor mediated entry of RSV is dependent of the cholesterol-rich lipid raft, what is the reason for the increased viral replication observed only relevant to the overexpression of receptor but not impacted by the reduced cholesterol-rich lipid raft from M CD treatment. Please explain it.

Page 13, lines 248-249, "Furthermore, treating the cells with M β CD led to an additional decrease in infectivity.", the treatment doesn't cause further decrease for siGF1R in RSVA2-infected HEp-2 and for siNCL and siGF1R in RSVB18537-infected HEp-2 from fig4G-H, which indicated the reduced NCL and IGF1R as well as the reduced lipid raft had no evident impact on virus entry, especially no additive impact from M β CD. Please explain it.

Page 13, lines 249-250, the results from both fig. 4G and H indicated again that the receptor-mediated entry of RSV seemed independent of the lipid raft. Please explain it.

Page 13, lines 250-251, from these results, it is impossible to draw this conclusion. Please explain it.

Page 24, line 446, it is not clear enough for the employed method here. please rewrite it. for example, what was the amount of the cells seeded on 24-well plates prior to the binding assays? was it necessary or not to wash cells for the binding assay as the internalization assay? for the internalization assay, where were these cells from? were they also from the same 24-well plate seeded cells? if and when were these cells pretreated with M β CD and infected with RSV? what did the drugs mean here, M β CD and Palivizumab, too? it is also necessary to introduce if all these experiments were done in duplicate or triple on 24-well plates?

Page 31, lines 606-607, please further introduce the detailed method to calculate the reduced CPE. How was the CPE developed before counting? If the CPE was stained before the observation under microscope?

Minor concerns:

Page 3, line 36, "RSV entry host cells involve generally different invasion pathways and are multistep processes."

Page 9, line 167, "150" ?

Page 12-13, lines 237-243, should be deleted

Page 17, line 327, "micropinocytosis" ?

Page 20, line 375, "with simvastatin" ?

Page 21, line 391, "g" ?

Page 23, line 430, please further provide the information such as the instrument and software etc. used to observe and count the cells stained by the FITC.

Page 40, figure 1, B and C are two different experiments, but they shared the same legend, which should be modified to match separate assay.

Page 43, figure 4, what do the vector and the vehicle mean, and how to be performed? and what are their roles? and it is reasonable to indicate the HEp-2 cells used in Fig4E and F are the established cell lines stably over-expressing NCL and IGF1R.

Page 45, figure 5 A, it is more reasonable to replace IFA determination to FFA, same on line 949, immunofluorescence should be replaced with FFA

FigS1, line 24, "After 22 hours", it seems this sentence should be modified as "After the culture media were replaced with those without the inhibitor or the control agent, the cultures were incubated another 22 hours and"

FigS2 , what does the Marker mean?

Dear Reviewer:

Thank you for your comments concerning our manuscript entitled “**Cholesterol-Rich Lipid Rafts Mediate Endocytosis as a Common Pathway for Respiratory Syncytial Virus Entry into Different Host Cells**” (Manuscript Number: **Spectrum01192-25**). These comments are all valuable and very helpful for revising and improving our manuscript. The detailed corrections are presented in a point-by-point response.

Reviewer #2 (Comments to the Author):

Major concerns:

1. Page 6, lines 93-94, "RSV infection causes 94 lung inflammation and asthma by cholesterol-dependent pathways (30-32)." no opinions or data presented in these refs demonstrated the lung inflammation and asthma caused by cholesterol-dependent pathways following RSV infection. For example, the opposite opinions, both enhancement and suppression of inflammation response from this pathway, were introduced in ref 30, no information mentioned on the cholesterol-dependent pathways following RSV infection in ref 31, and no direct evidence support this overexpression of NLRP3 inflammasome (critical in RSV-induced asthma) via this pathway following the elevated expression of orosomucoid 1-like protein 3 (ORMDL3) after RSV infection in ref 32. Please explain it.

Response: Thank you for pointing out these inappropriate references. As reported, RSV infection significantly increased the cellular cholesterol content (*Nat Commun.* PMID: 39060258)(1). RSV infection can induce the upregulation of ORMDL3 through histone hyperacetylation and subsequently promote NLRP3 inflammasome expression (*J Cell Physiol.* PMID: 37877592)(2). Some studies have also reported that the upregulation of inflammatory factors may be mediated by cholesterol-dependent pathways (*Nat Rev Immunol.* PMID: 25614320; *Sci Adv.* PMID: 36129973; *J Lipid Res.* PMID: 38522750 and *Nat Immunol.* PMID: 39838105)(3-6). As you mentioned, there is currently no direct evidence in the existing literature to suggest that lung inflammation and asthma are mediated by cholesterol-dependent pathways following RSV infection. Thus, we replaced and updated the original descriptions in the revised manuscript, as follows “Disruption of lipid raft has been shown to affect viral entry, including Marburg virus (MARV)(7), influenza A virus (IAV)(8), and SARS-CoV-2(9). Additionally, it has been reported that SARS-CoV-2 and IAV also depends on cholesterol for viral entry and attachment (4, 10).” (Lines 89-92, Page 5-6).

2. Page 12, lines 234-235, if receptor mediated entry of RSV is dependent of the cholesterol-rich lipid raft, what is the reason for the increased viral replication observed only relevant to the overexpression of receptor but not impacted by the

reduced cholesterol-rich lipid raft from M β CD treatment. Please explain it.

3. Page 13, lines 248-249, "Furthermore, treating the cells with M β CD led to an additional decrease in infectivity.", the treatment doesn't cause further decrease for siIGF1R in RSVA2-infected HEp-2 and for siNCL and siIGF1R in RSVB18537-infected HEp-2 from fig4G-H, which indicated the reduced NCL and IGFIR as well as the reduced lipid raft had no evident impact on virus entry, especially no additive impact from M β CD. Please explain it.

4. Page 13, lines 249-250, the results from both fig. 4G and H indicated again that the receptor-mediated entry of RSV seemed independent of the lipid raft. Please explain it.

Page 13, lines 250-251, from these results, it is impossible to draw this conclusion. Please explain it.

Response: Thank you for your professional comments. We sincerely apologize for these confusions that may have arisen regarding the interpretation of the manuscript, which was caused by an error in the relative quantification of the data presented in **Fig. 4E-H**. Each panel in **Fig. 4E-H** of the manuscript includes three data groups. The relative quantification was performed using the control within each group, resulting in inconsistencies between the quantified results and the conclusions drawn.

We have re-conducted relative quantification for **Fig. 4E-H** and updated the figures and added descriptions of relative quantitative calculations in the figure legends. In **Fig. 4E and F**, each dataset was normalized relative to the vector group within the untreated condition. The viral replication increased upon overexpression of the receptor, while it decreased when the cholesterol-rich lipid rafts were depleted by M β CD treatment. In **Fig. 4G and H**, each dataset was normalized relative to the siNC group within the DMSO-treated condition. The M β CD treatment causes further decrease for siIGF1R in RSV A2-infected HEp-2 cells and for siNCL and siIGF1R in RSV B18537-infected HEp-2 cells (**Fig. 4G and H**), but these decreases were slight. However, the restoration of NCL or IGF1R expression did not significantly increase infectivity, likely because M β CD disrupts lipid raft structures, preventing the normal recruitment of receptors to these rafts for utilization during RSV infection. We also made an explanation and updated the original descriptions in the revised manuscript (**Line 239-243, Page 12-13**). Overall, these results suggested that these receptors were involved in RSV entry into host cells via cholesterol-rich lipid raft-mediated endocytosis.

Figure 4. (**E, F**) Overexpression of NCL and IGF1R promoted RSV A2 or B18537 (MOI = 2) entry into cells. HEp-2 cells with stable NCL and IGF1R overexpression were pretreated with

M β CD (2 mM) for 1 hour, then infected with RSV A2 or B18537. Control cells were treated with an equivalent amount of DMSO alone. Infection rates were quantified using the fluorescence focus assay (FFA) assay at 24 h.p.i. The infection rate was calculated by determining the relative fluorescence intensity compared to the vector group within the untreated condition. Data are representative of three independent experiments and are presented as means \pm SD. Statistical differences were determined by Student's t-test. (G, H) Downregulation of NCL and IGF1R genes and M β CD reduced RSV entry into cells. NCL or IGF1R expression was knocked down by corresponding siRNA treatment. Some samples were pretreated with M β CD (2 mM) for 1 hour, while controls were exposed to DMSO alone. Cells were infected with RSV A2 or B18537(MOI = 2). Infection rates were quantified using the FFA assay at 24 h.p.i. The infection rate was calculated by determining the relative fluorescence intensity compared to the siNC group within the DMSO-treated condition. Data are representative of three independent experiments and are presented as means \pm SD. Statistical differences were determined by Student's t-test. * $P < 0.05$, ** $P < 0.01$, *** $P < 0.001$; n.s., not significant.

5. Page 24, line 446, it is not clear enough for the employed method here. please rewrite it. for example, what was the amount of the cells seeded on 24-well plates prior to the binding assays? was it necessary or not to wash cells for the binding assay as the internalization assay? for the internalization assay, where were these cells from? were they also from the same 24-well plate seeded cells? if and when were these cells pretreated with M β CD and infected with RSV? what did the drugs mean here, M β CD and Palivizumab, too? it is also necessary to introduce if all these experiments were done in duplicate or triple on 24-well plates?

Response: Thank you for your valuable feedback. We have thoroughly revised and enhanced the content of the experimental methods section in the revised manuscript. The details have been added, and the descriptive logic has been refined to ensure that the experimental procedures are presented with greater clarity, precision, and comprehensiveness. The drugs here were M β CD and palivizumab and we have replaced the description in the revised manuscript. The same treatment methods were used in the binding and internalization experiments of the two drugs, respectively. All these experiments were performed in triplicate on 24-well plates to ensure the reliability and reproducibility of the results (Lines 442-452, Page 23).

6. Page 31, lines 606-607, please further introduce the detailed method to calculate the reduced CPE. How was the CPE developed before counting? If the CPE was stained before the observation under microscope?

Response: Thank you for your comments. We utilized the Celigo Image Cytometer to quantify the CPE. The use of Celigo Image Cytometer for calculating CPE has been widely adopted in the literature (*J Fluoresc.* PMID: 37310590 and *Nat Commun.* PMID: 37990007)(11, 12). The Celigo software module "Confluence 1" was employed to determine the percentages of host cell confluence by analyzing the acquired bright-field images. The confluence percentages were calculated as the ratio

of the cell-covered surface area to the total surface area within the well, as directly measured by the image cytometer (**Lines 603-607, Page 30-31**).

Minor concerns:

1. Page 3, line 36, "RSV entry host cells involve generally different invasion pathways and are multistep processes."

Response: Thank you. RSV entry into host cells generally involves different invasive pathways including macropinocytosis and various endocytosis pathways. The entry process consists of two main steps: attachment of the virion to the host cell and fusion of the viral and host cell membranes.

2. Page 9, line 167, "150"?

Response: Thank you for pointing out the errors. We have thoroughly reviewed the experimental details and methods, and changed 150 min to 90 min to align consistently with the **Fig. 3A (Lines 165, Page 9)**.

3. Page 12-13, lines 237-243, should be deleted.

Response: Thank you. We have deleted the corresponding description.

4. Page 17, line 327, "micropinocytosis"?

Response: Thank you for pointing out the errors. We have corrected the "micropinocytosis" to "macropinocytosis" (**Lines 321, Page 16**).

5. Page 20, line 375, "with simvastatin"?

Response: Thank you. We have deleted "with simvastatin" in the revised manuscript (**Lines 369, Page 19**).

6. Page 21, line 391, "g"?

Response: Thank you. We have corrected "g" to italic in the revised manuscript (**Lines 385, Page 20**).

7. Page 23, line 430, please further provide the information such as the instrument and software etc. used to observe and count the cells stained by the FITC.

Response: Thank you. We have revised this part of the content in the revised manuscript, as follows: "RSV-FITC was observed using the PerkinElmer Operetta

CLS. Fluorescence intensity quantification of RSV-FITC using ImageJ” (Lines 423-425, Page 22).

8. Page 40, figure 1, B and C are two different experiments, but they shared the same legend, which should be modified to match separate assay.

Response: Thank you for your suggestions. We have revised this part of the content in the revised manuscript, as follows: “(B) Cholesterol depletion had no effect on RSV binding. Cells were preincubated in a medium containing M β CD (5 mM) at 37°C for 1 hour. Subsequently, the cells were infected with RSV A2, B18537, or ON1 (MOI = 10) on ice for 1 hour, respectively. Palivizumab (10 μ g/mL) was used as a control. After five cycles of washing, cells were collected, and the viral RNA was detected. (C) Cholesterol depletion reduced the internalization of RSV into cells. After viral binding, the cells were added to the medium supplemented with M β CD (5 mM) or palivizumab (10 μ g/mL), and then transferred to 37°C incubator for 1 hour. Thereafter, the cells were frozen on ice and then treated with proteinase K (500 ng/mL) on ice for 1 hour. After five cycles of washing, cells were collected, and the viral RNA was detected.” (Lines 880-887, Page 39).

9. Page 43, figure 4, what do the vector and the vehicle mean, and how to be performed? and what are their roles? and it is reasonable to indicate the HEP-2 cells used in Fig4E and F are the established cell lines stably over-expressing NCL and IGFIR.

Response: Thank you for your comments. The "vector" refers to plasmid vectors used for stable overexpression of target genes (NCL or IGF1R) in HEP-2 cells. The empty pcDNA3.1 vector was used as a negative control for overexpression of NCL and IGF1R to exclude nonspecific effects of the vector itself. The “vehicle” refers to the untreated group and was used as a negative control for DMSO and M β CD-treated groups. We have corrected the “vehicle” to “untreatment” (Fig. 4E and F). Additionally, we also have indicated the HEP-2 cells used in Fig. 4E and F are the established cell lines stably over-expressing NCL and IGFIR in the revised manuscript (Lines 230, Page12 and Lines 934-935, Page 42).

10. Page 45, figure 5 A, it is more reasonable to replace IFA determination to FFA, same on line 949, immunofluorescence should be replaced with FFA

Response: Thank you. We have corrected the IFA determination to FFA (Lines 960, Page 44).

11. FigS1, line 24, "After 22 hours", it seems this sentence should be modified as "After the culture media were replaced with those without the inhibitor or the control agent, the cultures were incubated another 22 hours and"

Response: Thank you. We have revised the description accordingly in the revised supplementary materials (**Fig. S1, Line 24-26**).

12. FigS2 , what does the Marker mean?

Response: Thanks. Marker was defined as Caveolin-1 or Dynamin-2, now we have labeled clearly marker in **Fig. S2** in the revised manuscript.

REFERENCE

1. Chen L, Zhang J, Xu W, Chen J, Tang Y, Xiong S, Li Y, Zhang H, Li M, Liu Z. 2024. Cholesterol-rich lysosomes induced by respiratory syncytial virus promote viral replication by blocking autophagy flux. *Nat Commun* 15:6311.
2. Cheng Q, He F, Zhao W, Xu X, Shang Y, Huang W. 2023. Histone acetylation regulates ORMDL3 expression-mediated NLRP3 inflammasome overexpression during RSV-allergic exacerbation mice. *J Cell Physiol* 238:2904-2923.
3. Tall AR, Yvan-Charvet L. 2015. Cholesterol, inflammation and innate immunity. *Nat Rev Immunol* 15:104-16.
4. Gao P, Ji M, Liu X, Chen X, Liu H, Li S, Jia B, Li C, Ren L, Zhao X, Wang Q, Bi Y, Tan X, Hou B, Zhou X, Tan W, Deng T, Wang J, Gao GF, Zhang F. 2022. Apolipoprotein E mediates cell resistance to influenza virus infection. *Sci Adv* 8:eabm6668.
5. Yalcinkaya M, Liu W, Xiao T, Abramowicz S, Wang R, Wang N, Westerterp M, Tall AR. 2024. Cholesterol trafficking to the ER leads to the activation of CaMKII/JNK/NLRP3 and promotes atherosclerosis. *J Lipid Res* 65:100534.
6. Belabed M, Park MD, Blouin CM, Balan S, Moon CY, Freed G, Quijada-Álamo M, Peros A, Mattiuz R, Reid AM, Yatim N, Boumelha J, Azimi CS, LaMarche NM, Troncoso L, Amabile A, Le Berichel J, Chen ST, Wilk CM, Brown BD, Radford KJ, Ghosh S, Rothlin CV, Yvan-Charvet L, Marron TU, Puleston DJ, Wagenblast E, Bhardwaj N, Lamaze C, Merad M. 2025. Cholesterol mobilization regulates dendritic cell maturation and the immunogenic response to cancer. *Nat Immunol* 26:188-199.
7. Bavari S, Bosio CM, Wiegand E, Ruthel G, Will AB, Geisbert TW, Hevey M, Schmaljohn C, Schmaljohn A, Aman MJ. 2002. Lipid raft microdomains: a gateway for compartmentalized trafficking of Ebola and Marburg viruses. *J Exp Med* 195:593-602.
8. Verma DK, Gupta D, Lal SK. 2018. Host Lipid Rafts Play a Major Role in Binding and Endocytosis of Influenza A Virus. *Viruses* 10.
9. Teixeira L, Temerozo JR, Pereira-Dutra FS, Ferreira AC, Mattos M, Goncalves BS, Sacramento CQ, Palhinha L, Cunha-Fernandes T, Dias SSG, Soares VC, Barreto EA, Cesar-Silva D, Fintelman-Rodrigues N, Pao CRR, de Freitas CS, Reis PA, Hottz ED, Bozza FA, Bou-Habib DC, Saraiva EM, de Almeida CJG, Viola JPB, Souza TML, Bozza PT. 2022. Simvastatin Downregulates the SARS-CoV-2-Induced Inflammatory Response and Impairs Viral Infection Through Disruption of Lipid Rafts. *Front Immunol* 13:820131.
10. Sanders DW, Jumper CC, Ackerman PJ, Bracha D, Donlic A, Kim H, Kenney D, Castello-Serrano I, Suzuki S, Tamura T, Tavares AH, Saeed M, Holehouse AS, Ploss A, Levental I, Douam F, Padera RF, Levy BD, Brangwynne CP. 2021. SARS-CoV-2 requires cholesterol for viral entry and pathological syncytia formation. *Elife* 10.
11. Wang H, Yang Q, Liu X, Xu Z, Shao M, Li D, Duan Y, Tang J, Yu X, Zhang Y, Hao A, Wang Y, Chen J, Zhu C, Guddat L, Chen H, Zhang L, Chen X, Jiang B, Sun L, Rao Z, Yang H. 2023. Structure-based discovery of dual pathway inhibitors for SARS-CoV-2 entry. *Nat Commun* 14:7574.
12. St Clair LA, Chan LL, Boretsky A, Lin B, Spedding M, Perera R. 2024. High-Throughput SARS-CoV-2 Antiviral Testing Method Using the Celigo Image Cytometer. *J Fluoresc* 34:561-570.

Re: Spectrum01192-25R1 (**Cholesterol-Rich Lipid Rafts Mediate Endocytosis as a Common Pathway for Respiratory Syncytial Virus Entry into Different Host Cells**)

Dear Prof. Xinwen Chen:

Thank you for the privilege of reviewing your work. Below you will find my comments, instructions from the Spectrum editorial office, and the reviewer comments.

Revision Guidelines

Sincerely,
Leiliang Zhang
Editor
Microbiology Spectrum

Reviewer #2 (Comments for the Author):

minor concern:

please include the method for restoration or recover experiment, performed in Fig4G and H, in the corresponding method section and the legend.

Dear Reviewer:

Thank you for your comments concerning our manuscript entitled “**Cholesterol-Rich Lipid Rafts Mediate Endocytosis as a Common Pathway for Respiratory Syncytial Virus Entry into Different Host Cells**” (Manuscript Number: **Spectrum01192-25R1**). These constructive comments are greatly appreciated and have helped strengthen the manuscript. Our point-by-point responses to each suggestion are provided in the sections that follow.

Reviewer #2 (Comments to the Author):

Minor concern:

Please include the method for restoration or recover experiment, performed in Fig4G and H, in the corresponding method section and the legend.

Response: Thank you for your suggestions. We have revised this part of the legend in the revised manuscript, as follows: “One group of cells was transfected with specific siRNAs designed to target either NCL or IGF1R for gene knockdown. A second group underwent siRNA-mediated knockdown of NCL or IGF1R, subsequently followed by transfection with plasmids expressing NCL or IGF1R (500 ng/well) to restore their expression levels. Some samples were pretreated with M β CD (2 mM) for 1 hour, while controls were exposed to DMSO alone. Cells were infected with RSV A2 or B18537 (MOI = 2) combined with drug treatment for 2 hours. Infection rates were quantified using the FFA assay at 24 h.p.i.” (**Lines 957-964, Page 44**).

And the materials and methods related to Fig. 4G and H have been incorporated into the revised manuscript, as follows: “**siRNA knockdown and plasmid rescue.** HEp-2 cells, at a density of 1×10^5 cells per well, were cultured in 24-well plates for a duration of 20 hours. Subsequently, the cells underwent transfection with siRNA targeting either NCL or IGF1R at a concentration of 40 nM to induce gene knockdown. A subset of these cells was further subjected to transfection with plasmids expressing NCL or IGF1R (500 ng/well) to facilitate functional rescue. After these interventions, selected samples from both groups were pretreated with 2 mM M β CD for 1 hour, whereas control samples received a vehicle treatment consisting of 0.1% DMSO. Following these treatments, all cells were infected with RSV A2 or B18537 (MOI = 2) for 2 hours in the medium that also contained 2 mM M β CD. The tissue culture medium without inhibitor was replaced after 2 hours of incubation, and 24 h.p.i., the infection rate was quantified using an FFA assay. This involved fixing the cells, immunostaining with RSV-FITC antibody, and analyzing the relative fluorescence intensity, which was normalized to the DMSO-treated siNC control group, set as 100% infection. The data are presented as means \pm SD from three independent experimental replicates.” (**Lines 508-515, Page 26**).

Re: Spectrum01192-25R2 (**Cholesterol-Rich Lipid Rafts Mediate Endocytosis as a Common Pathway for Respiratory Syncytial Virus Entry into Different Host Cells**)

Dear Prof. Xinwen Chen:

Your manuscript has been accepted, and I am forwarding it to the ASM production staff for publication. Your paper will first be checked to make sure all elements meet the technical requirements. ASM staff will contact you if anything needs to be revised before copyediting and production can begin. Otherwise, you will be notified when your proofs are ready to be viewed.

Sincerely,
Leiliang Zhang
Editor
Microbiology Spectrum